# Bypassing spike sorting: Density-based decoding using spike localization from dense multielectrode probes

**Yizi Zhang**[1*]    **Tianxiao He**[1,2*]    **Julien Boussard**[1]    **Charlie Windolf**[1]    **Olivier Winter**[3]
**Eric Trautmann**[1]    **Noam Roth**[4]    **Hailey Barrell**[4]    **Mark Churchland**[1]    **Nicholas A. Steinmetz**[4]
**The International Brain Laboratory**[3]    **Erdem Varol**[2]    **Cole Hurwitz**[1]    **Liam Paninski**[1]

[1]Columbia University    [2]New York University    [3]The International Brain Laboratory
[4] University of Washington in Seattle

## Abstract

Neural decoding and its applications to brain computer interfaces (BCI) are essential for understanding the association between neural activity and behavior. A prerequisite for many decoding approaches is *spike sorting*, the assignment of action potentials (spikes) to individual neurons. Current spike sorting algorithms, however, can be inaccurate and do not properly model uncertainty of spike assignments, therefore discarding information that could potentially improve decoding performance. Recent advances in high-density probes (e.g., Neuropixels) and computational methods now allow for extracting a rich set of spike features from unsorted data; these features can in turn be used to directly decode behavioral correlates. To this end, we propose a spike sorting-free decoding method that directly models the distribution of extracted spike features using a mixture of Gaussians (MoG) encoding the uncertainty of spike assignments, without aiming to solve the spike clustering problem explicitly. We allow the mixing proportion of the MoG to change over time in response to the behavior and develop variational inference methods to fit the resulting model and to perform decoding. We benchmark our method with an extensive suite of recordings from different animals and probe geometries, demonstrating that our proposed decoder can consistently outperform current methods based on thresholding (i.e. multi-unit activity) and spike sorting. Open source code is available at https://github.com/yzhang511/density_decoding.

## 1   Introduction

Decoding methods for large-scale neural recordings are opening up new ways to understand the neural mechanisms underlying cognition and behavior in diverse species (Urai et al., 2022). The emergence of high-density multi-electrode array (HD-MEA) devices introduced a tremendous increase in the number of extracellular channels that can be recorded simultaneously (Jun et al., 2017; Steinmetz et al., 2021), leading to scalable and high-bandwith brain computer interfaces (BCI) systems (Musk et al., 2019; Paulk et al., 2022).

Traditional neural decoding methods assume that spiking activity has already been correctly spike-sorted. As a result, these methods are not appropriate for situations where sorting cannot be performed with high precision. Despite intensive efforts towards automation, current spike sorting algorithms still require manual supervision to ensure sorting quality (Steinmetz et al., 2018). Even after careful curation, current spike sorters suffer from many sources of errors including erroneous spike assignment (Deng et al., 2015). The dense spatial resolution of HD probes makes some known issues of spike sorting even more evident. With the increased density of the recording channels, the

37th Conference on Neural Information Processing Systems (NeurIPS 2023).

probability of visibly overlapping spikes (spike collisions) is higher (Buccino et al., 2022). Even for the same HD dataset, different spike sorters have low agreement on the isolated units and can find a significant number of poorly sorted and noisy units (Buccino et al., 2020). Consequently, only single units that pass quality control metrics are included in many neural coding studies (IBL et al., 2022).

Because the spike-sorting problem remains unresolved, alternative approaches that do not rely on sorted single-units for decoding have been proposed. A popular choice is multi-unit threshold crossing that uses spiking activity on each electrode for decoding (Fraser et al., 2009; Trautmann et al., 2019). However, this approach ignores the fact that the signal on each electrode is a combination of signals from different neurons, thus making inefficient use of the data (Todorova et al., 2014). Ventura (2008) proposed a spike-sorting free decoding paradigm that estimates neuronal tuning curves from electrode tuning curves and then infers the behavior of interest using the estimated tuning curves and newly observed electrode spike trains. More recently, Chen et al. (2012); Kloosterman et al. (2014); Deng et al. (2015); Rezaei et al. (2021) developed spike feature decoding methods that use marked point processes to characterize the relationship between the behavior variable and features of unsorted spike waveforms. However, these decoders based on state-space models make explicit assumptions about the underlying system dynamics which reduce their flexibility in capturing complex relationships in the data. Moreover, these methods mainly utilize simple waveform features for decoding such as the maximum amplitude on each electrode and do not take advantage of HD spike features such as the estimated spike location.

To leverage the spatial spread and density of HD probes, Hurwitz et al. (2019); Boussard et al. (2021) developed spike localization methods. These methods estimate the source location of a detected spike; this is a low-dimensional feature that is informative about the firing neuron's identity. We propose a probabilistic model-based decoding method that scales to HD-MEA devices and utilizes these novel localization features in conjunction with additional waveform features. We use a mixture of Gaussians (MoG) model to encode the uncertainty associated with spike assignments in the form of parametric distributions of the spike features. Unlike traditional MoG models with a fixed mixing proportion, our method allows the mixing proportion to depend on the behavior of interest and change over time. This is motivated by the theory that behavioral covariates that modulate neurons' firing rates also contain information about spike identities and that such tuning information should be incorporated into spike sorting and neural decoding in order to obtain unbiased and consistent tuning function estimates (Ventura, 2009). To infer the functional relationship between spike features and behavioral correlates, we employ automatic differentiation variational inference (ADVI) (Kucukelbir et al., 2017) and coordinate ascent variational inference (CAVI) (Blei et al., 2017), which enable us to perform efficient and accurate inference while considering the behavior-modulated MoG model.

We apply our method to a large number of HD recordings and decode various types of behavioral correlates. Experimental results show that our decoder consistently outperforms decoders based on multi-unit threshold crossings and single-units sorted by Kilosort 2.5 (Pachitariu et al., 2023). We further validate the robustness of our method by applying it to recordings with different levels of sorting quality, HD probes with varying geometry, and recordings from multiple animal species. Consistent with previous results, our findings indicate that relying solely on "good" units, as determined by sorting quality metrics, leads to information loss and suboptimal decoding performance. This observation motivates our transition to a spike sorting-free decoding framework which enables us to extract more information from the spiking activity and improve decoding performance.

## 2 Method

Consider an electrophysiological recording comprised of $K$ trials, where each trial is divided into $T$ equally spaced time bins. Let $\{s_{itk}\}_{i=1}^{n_{tk}}, s_{itk} \in \mathbb{R}^D$ denote a set of spike features, where $i$ indexes the $i$-th spike, $n_{tk}$ represents the number of spikes collected in the $t$-th time bin of the $k$-th trial, and $D$ is the dimension of spike features. For example, the spike feature $s_{itk} = (x_{itk}, z_{itk}, a_{itk}) \in \mathbb{R}^3$ includes the spike location along the x- and z-axis of the probe, and its maximum peak-to-peak (max ptp) amplitude. Let $y_k \in \mathbb{R}^T$ be the observed time-varying behavior in the trial $k$, e.g., the speed of a rotating wheel controlled by a mouse. When the behavior in the trial $k$ does not vary over time, it can take on either a binary ($y_k \in \{0, 1\}$) or scalar ($y_k \in \mathbb{R}$) value, e.g., the mouse responds to a

scalar-valued stimulus by making a binary decision.

The proposed decoding method comprises an encoder and a decoder model. During the training of the model, the encoder captures the relationship between the observed spike feature distribution and the observed behavior. During the testing phase, the decoder utilizes the learned relationship from the encoder to predict the unobserved behavior based on newly observed spike features. In the following section, we present a general formulation of the encoding-decoding paradigm and provide more detailed implementations in the supplementary materials.

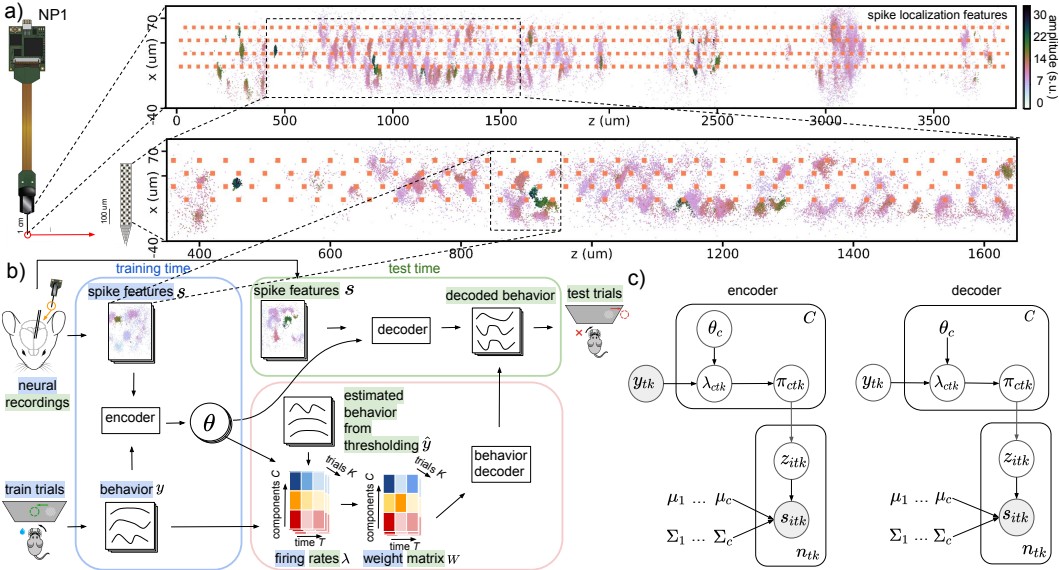

Figure 1: **Decoding paradigm and graphical model.** (a) Spike localization features, $(x, z)$, the locations of spikes along the width and depth of the NP1 probe, and waveform features, $a$, the maximum peak-to-peak (max ptp) amplitudes of spikes. Amplitude is measured in standard units (s.u.). Spike features from the entire probe are shown, and we focus on a specific segment of the probe. (b) During the training phase, the encoder takes the observed spike features $s$ and behavior $y$ from the train trials as inputs and then outputs the variational parameters $\theta$ which control the dependence of the firing rate $\lambda$ on the behavior $y$. At test time, the decoder utilizes the learned model parameters $\theta$ obtained from the encoder and the observed spike features $s$ from the test trials to predict the corresponding behavior in the test trials. To ensure reliable decoding of behaviors, we initially calculate the $\lambda$ during training using the learned $\theta$ and observed behaviors $y$ from the train trials. Then, we compute the $\lambda$ during test time using the learned $\theta$ and the estimated behavior $\hat{y}$ obtained through multi-unit thresholding from the test trials. Finally, we generate the weight matrix $W$ for both the train and test trials as input to the final behavior decoder, e.g., linear regression or neural networks (Glaser et al., 2020; Livezey and Glaser, 2021). (c) In the encoder, the firing rates of each MoG component $\lambda_{ctk}$ are modulated by the observed behavior $y_{tk}$ in the train trials. This modulation affects the MoG mixing proportion $\pi_{ctk}$, which in turn determines the spike assignment $z_{itk}$ that generates the observed spike features $s_{itk}$ in the train trials. In the decoder, the behavior $y_{tk}$ in the test trials is unknown and considered as a latent variable to be inferred. The decoder uses the observed spike features $s_{itk}$ from the test trials along with the fixed model parameters $\theta_c$ learned by the encoder to infer the latent behavior $y_{tk}$.

**Encoder.** The multivariate spike feature distribution is modeled using a mixture of Gaussian (MoG). The encoder generative model is as follows:

$$\lambda_{ctk} = \lambda(t, y_k, \theta_c), \ \theta_c \sim p(\theta_c), \tag{1}$$

$$z_{itk} \sim \text{Categorical}(z_{itk}; \pi_{tk}), \ \pi_{tk} = \{\pi_{ctk}\}_{c=1}^{C}, \ \pi_{ctk} = \frac{\lambda_{ctk}}{\sum_{c'} \lambda_{c'tk}}, \tag{2}$$

$$s_{itk} \sim \mathcal{N}(s_{itk}; \eta_{z_{itk}}), \ \eta_c = (\mu_c, \Sigma_c), \tag{3}$$

where $\lambda(\cdot)$ is a function that describes the firing rate's dependence on behaviors $y_k$, while $p(\theta_c)$ represents a general prior on $\theta_c$ encompassing the model parameters for the mixture component $c$. Intuitively, the behavior-dependent $\lambda$ governs the mixing proportion $\pi$ of the MoG, which determines the specific component $c$ from which a spike feature $s$ is generated. As $\lambda$ varies over time in response to $y$, spikes that are spatially close and share similar waveform features may originate from different MoG components at different time points within a trial. In our implementation, we parameterize $\lambda$ using a generalized linear model (GLM), but alternative models such as neural networks (NN) can also be used; see the supplementary material for the GLM configuration.

We employ variational inference (VI) to learn the unknown quantities. In the standard MoG setting (Blei et al., 2017), our goal is to infer the spike assignment $z$ which indicates the latent component from which the observation $s$ originates. However, unlike the standard MoG, our spike assignment $z$ is influenced by the firing rates $\lambda$ of the neurons which are modulated by the behavior $y$. Consequently, learning the association between $\lambda$ and $y$ necessitates the estimation of the unknown model parameters $\theta$. Our objective is to simultaneously learn both the latent variables $z$ and model parameters $\theta$ based on the observed spike features $s$ and behavior $y$. To accomplish this, we posit a mean-field Gaussian variational approximation

$$q(z, \theta) = \prod_{c,t} q(z_{ct})q(\theta_c) \tag{4}$$

for the posterior $p(z, \theta \mid s, y)$. Subsequently, we employ the CAVI or ADVI methods to maximize the evidence lower bound (ELBO) and compute updates for $z$ and $\theta$. Analogous to the standard Expectation-Maximization (EM) algorithm for MoG, the proposed CAVI and ADVI procedures consist of an E step and a M step. The E step, which updates $z$, closely resembles that of the ordinary MoG, while the M step, responsible for updating $\theta$, differs. CAVI utilizes coordinate ascent to find $\theta$ that maximizes the ELBO, while ADVI employs stochastic gradient ascent for $\theta$ updates. For detailed information on the CAVI and ADVI model specifications, refer to Supplement 1 and 2.

**Decoder.** The decoder adopts the same generative model in Equation 1-3 as the encoder with two distinctions: 1) $y_k$ is unobserved and considered a latent variable we aim to estimate, i.e., $y_k \sim p(y_k)$, where $p(y_k)$ is a general prior, and 2) the model parameters $\theta$ are obtained from the encoder and kept constant. In practice, the choice of prior relies on the nature of $y_k$. For instance, if $y_k$ is binary, we can sample from a Bernoulli distribution while a Gaussian process prior can capture the temporal correlation between time steps if $y_k \in \mathbb{R}^T$. The posterior $p(z, y \mid s)$ is approximated using a mean-field Gaussian variational approach

$$q(z, y) = \prod_{c,t} q(z_{ct})q(y). \tag{5}$$

We employ standard CAVI or ADVI methods to infer $z$ and decode $y$.

**Robust behavior decoding.** In practice, we found that direct decoding of $y$ using the approximated posterior $q(y)$ in Equation 5 was not robust, leading to decoding results of inconsistent quality across different datasets. Although the factorization described in Equation 5 may not fully exploit the available information for predicting $y$, it is useful for learning about the spike assignment $z$. To enhance decoding robustness, we compute a weight matrix from the MoG outputs as input to the final behavior decoder. The weight matrix, denoted as $W$, has dimensions $K \times C \times T$ and entries $W_{kct} := \sum_{i=1}^{n_{tk}} q(z_{ikct})$, which capture the posterior probability of assigning spike $i$ collected at time $t$ of the trial $k$ into the component $c$. In scenarios such as spike sorting or multi-unit thresholding, spikes are assigned deterministically to one of $C$ sorted single units or thresholded channels. In this case, $W$ has one-hot rows, and each entry $W_{kct}$ represents the number of spikes belonging to trial

$k$, time $t$ and unit (channel) $c$. To obtain $q(z_{ikct})$, we rely on estimating the posterior $\pi_{ctk}$, which requires either the observed $y_{tk}$ or the estimated $\hat{y}_{tk}$. During model training, we can substitute the observed $y_{tk}$ into Equation 1 to calculate the posterior $\pi_{ctk}$ for the train trials. At test time, we use the estimated $\hat{y}_{tk}$ obtained from multi-unit thresholding along with the learned encoder parameters $\theta$ to calculate the posterior $\pi_{ctk}$ for the test trials. With the posterior $\pi_{ctk}$ in hand, we then estimate $q(z)$ to compute the weight matrix $W$ for both the train and test trials, which serves as input to the final behavior decoder. The choice of the behavior decoder depends on the user's preference. For instance, we can use logistic regression as the behavior decoder for binary $y_k$, and ridge regression for $y_k \in \mathbb{R}^T$ that exhibit temporal variations. Additional information regarding the selection of the behavior decoder can be found in Supplement 5.

## 3 Experiments

We conducted experiments using both electrophysiological and behavior data obtained from the International Brain Laboratory (IBL) (IBL et al., 2021). The electrophysiological recordings were acquired using Neuropixels (NP) probes that were implanted in mice performing a decision-making task. Each recording comprises multiple trials with several behavioral variables recorded during each trial including the choice, face motion energy, and wheel speed; see Figure 2 (a) for details. Each trial has a duration of 1.5 seconds and is divided into 30 time bins of 50 milliseconds length. The NP probe spans multiple brain regions; an example of the brain parcellation can be seen in Figure 3. To prepare the recordings for decoding, we first applied IBL's standard destriping procedure (Chapuis et al., 2022) to reduce artifacts. Then, we used a subtraction-based spike detection and denoising method described in Boussard et al. (2023). After preprocessing, we computed the spike locations (Boussard et al., 2021) to acquire spike features for decoding and then utilized registration techniques (Windolf et al., 2022) to correct for motion drift in the recorded data. Further details about data preprocessing can be found in Supplement 4. In all experiments, we used spike locations along the width and depth of the NP probe, and maximum peak-to-peak amplitudes of spikes for decoding. We selected this set of spike features based on empirical evidence from our experiments, which showed their good decoding performance. Furthermore, previous studies (Boussard et al., 2023; Hilgen et al., 2017) have also recognized that these features were highly informative about unit identity. Figure 1(a) illustrates the spike localization and waveform features that were utilized for decoding.

We evaluate the performance of our decoding method by comparing it to the following baselines: (1) Spike-thresholded decoders which utilize the spiking activity on each electrode after a voltage-based detection step. (2) Spike-sorted decoders that utilize all single-units found using Kilosort (KS) 2.5 (Pachitariu et al., 2016). (3) Spike-sorted decoders based on "good" units which consist of KS units that have passed IBL's quality control procedure (IBL et al., 2022). The parameters used for KS were tuned across multiple IBL datasets as described by IBL's spike sorting white paper (Chapuis et al., 2022).

To assess the quality of decoding, we perform 5-fold cross validation (CV) and compute relevant decoding metrics. The coefficient of determination ($R^2$) is used to evaluate continuous behavior decoding (e.g., motion energy and wheel speed) while accuracy is used for discrete behaviors (e.g., choice). To demonstrate the efficacy of our approach in a wide range of settings, we conduct the following experiments.

**Varying levels of spike sorting quality.**    Our objective is to compare the proposed decoding method to spike-sorted decoders using datasets which have varying levels of spike sorting quality. We apply our method to two datasets with high sorting quality ("good" sorting) and two with low sorting quality ("bad" sorting). The quality of sorting is assessed using IBL's quality metrics (Chapuis et al., 2022). Although motion registration has been performed, we find that the recordings that have "bad" sortings are more affected by motion drift then the recordings which have "good" sortings; see supplementary materials.

**Different brain regions from 20 datasets.**    To demonstrate the efficacy of our method across many different datasets, we decode 20 IBL datasets (IBL et al., 2022). In these datasets, mice perform a behavioral task while NP1 probes record activity from multiple brain regions. These brain regions

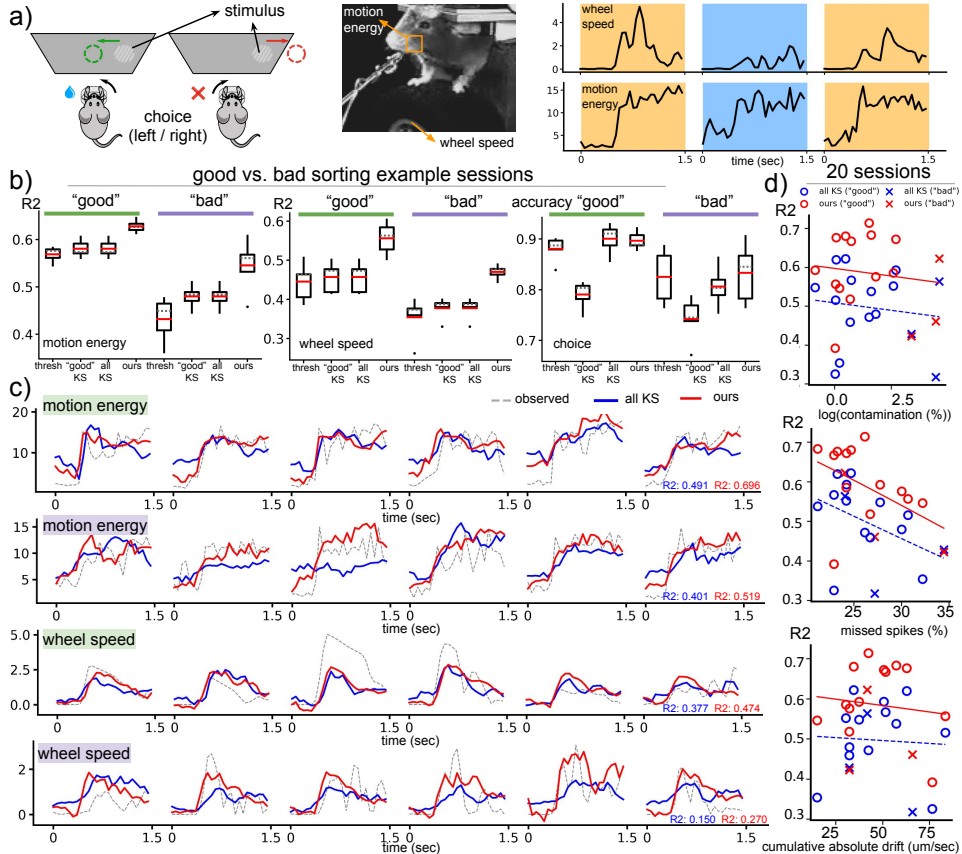

Figure 2: **Density-based decoding is robust to varying levels of spike sorting quality.** (a) We decode various behaviors including choice, motion energy and wheel speed. In the experimental setup, the mouse detects the presence of a visual stimulus to their left or right and indicates the perceived location (choice) by turning a steering wheel in the corresponding direction. Motion energy is calculated within a square region centered around the mouse's whiskers. The example behavior traces are distinguished by different colors for each trial. (b) We compare decoders using two experimental sessions with "good" sorting quality (represented by the color green) and two sessions with "bad" sorting quality (represented by the color purple) based on IBL's quality metrics. The box plots display various statistical measures including the minimum, maximum, first and third quartiles, median (indicated by a gray dashed line), mean (indicated by a red solid line), and outliers (represented by dots). These decoding metrics are obtained from a 5-fold CV and are averaged across both "good" and "bad" sorting example sessions. (c) We compare the traces decoded by spike-sorted decoders and our method on example sessions with "good" sorting quality (indicated by green) and "bad" sorting quality (indicated by purple). (d) The scatter plots depict the decoding quality of motion energy, measured by $R^2$, with respect to various spike-sorting quality metrics. Each point represents one of the 20 IBL sessions, and different colors and shapes are used to distinguish between the type of decoder and sorting quality. The sorting quality metrics include "contamination," which estimates the fraction of unit contamination (Hill et al., 2011), "drift," which measures the absolute value of the cumulative position change in micrometers per second (um/sec) of a given KS unit, and "missed spikes," which approximates the fraction of missing spikes from a given KS unit (Hill et al., 2011). These metrics are averaged across all KS units in a session. The scatter plots demonstrate that decoding quality tends to decrease when sorting quality is compromised. However, our method outperforms spike-sorted decoders even in the presence of these sorting issues.

are repeatedly targeted across all the datasets. To explore how behaviors are linked to specific brain regions, we use spikes that are confined to a particular area of the mouse brain for decoding. We collect decoding results from the posterior thalamic nucleus (PO), the lateral posterior nucleus (LP), the dentate gyrus (DG), the cornu ammonis (CA1) and the anterior visual area of the visual cortex (VISa).

**Different probe geometry.** Our method is capable of decoding electrophysiological data from a variety of HD probes. To demonstrate this, we apply our method on Neuropixels 2.4 (NP2.4) and Neuropixels 1.0 in nonhuman primates (NP1-NHP) datasets. The NP2.4 and NP1-NHP recordings are preprocessed using an identical pipeline as employed for NP1; see Supplement 4 for details. For spike sorting, the NP2.4 and NP1 adopt the same KS parameters as outlined in IBL's spike sorting procedure. Different KS parameters are utilized for NP1-NHP probes which are detailed in Trautmann et al. (2023).

- **NP2.4:** Each NP2.4 probe consists of four shanks and a total of 384 channels (Steinmetz et al., 2021). NP2.4 probes are more dense (i.e., more channels in a given area) than NP1. The mice were trained in accordance with the IBL experiment protocols to perform a visual decision-making task. The behavioral correlates we decode are choice, motion energy, and wheel speed.
- **NP1-NHP:** NP1-NHP is designed for nonhuman primate species, such as macaques. The NP1-NHP probe maintains the same number of channels as NP1 (384), but its overall length is extended, resulting in a sparser configuration compared to NP1 (Trautmann et al., 2023). During the experiment, the macaque underwent training in a sequential multi-target reaching task (Marshall et al., 2022). The behavioral correlate we decode is the monkey's arm force. The probe was implanted in the macaque's motor cortex.

**Comparison to a state-of-the-art clusterless decoder.** Although the lack of available code for prior methods make comprehensive comparisons difficult, we benchmark our density-based decoder against a state-of-the-art clusterless decoder on datasets from both HD probes and multiple tetrodes. We compare our method to the clusterless point process decoder of Denovellis et al. (2021), which utilizes a marked point process to connect spike features with behaviors. For more details of this comparison, see Section 9 of the supplementary materials.

To decode the binary choice variable, we utilize the CAVI algorithm described in Supplement 2. For continuous behaviors like motion energy, wheel speed, and arm force, we employ the ADVI algorithm outlined in Supplement 1. We specify the maximum number of iterations as a hyperparameter in the CAVI model as it requires analytical updates for the model parameters. Running the CAVI encoder and decoder for fewer than 50 iterations yields satisfactory decoding outcomes. As for the ADVI algorithm, we implement it in PyTorch and update the model parameters using the Adam optimizer with a learning rate of 0.001 and a batch size of 6. The ADVI model is run for 1000 iterations. Open source code is available at https://github.com/yzhang511/density_decoding.

## 4 Results

**Varying levels of spike sorting quality.** The performance of our method in comparison to the spike-thresholded and spike-sorted decoders for both the "good" and "bad" sorting examples is summarized in Figure 2. For the "good" sorting examples, our method has the highest decoding performance for motion energy and wheel speed. For choice decoding, our approach is comparable to decoding based on all KS single-units and better than decoders based on multi-unit thresholding and "good" KS units. For the "bad" sorting sessions, the gap in decoding performance between our method and other decoders is more pronounced. Example traces are illustrated in Figure 2 (c), which demonstrate that the behavior traces decoded by our method closely match the observed traces compared to decoded traces from the sorted decoders. In Figure 2 (d), we quantify the relationship between sorting quality and decoding quality using data from 20 IBL sessions. For all three quality metrics, the performance of our decoder and the spike-sorted decoder decreases as the quality of the sorting decreases. Despite this decrease in performance, our method consistently has better performance than the spike-sorted decoder even in the presence of significant motion drift as well as a when there is a large fraction of missed spikes or contaminated units.

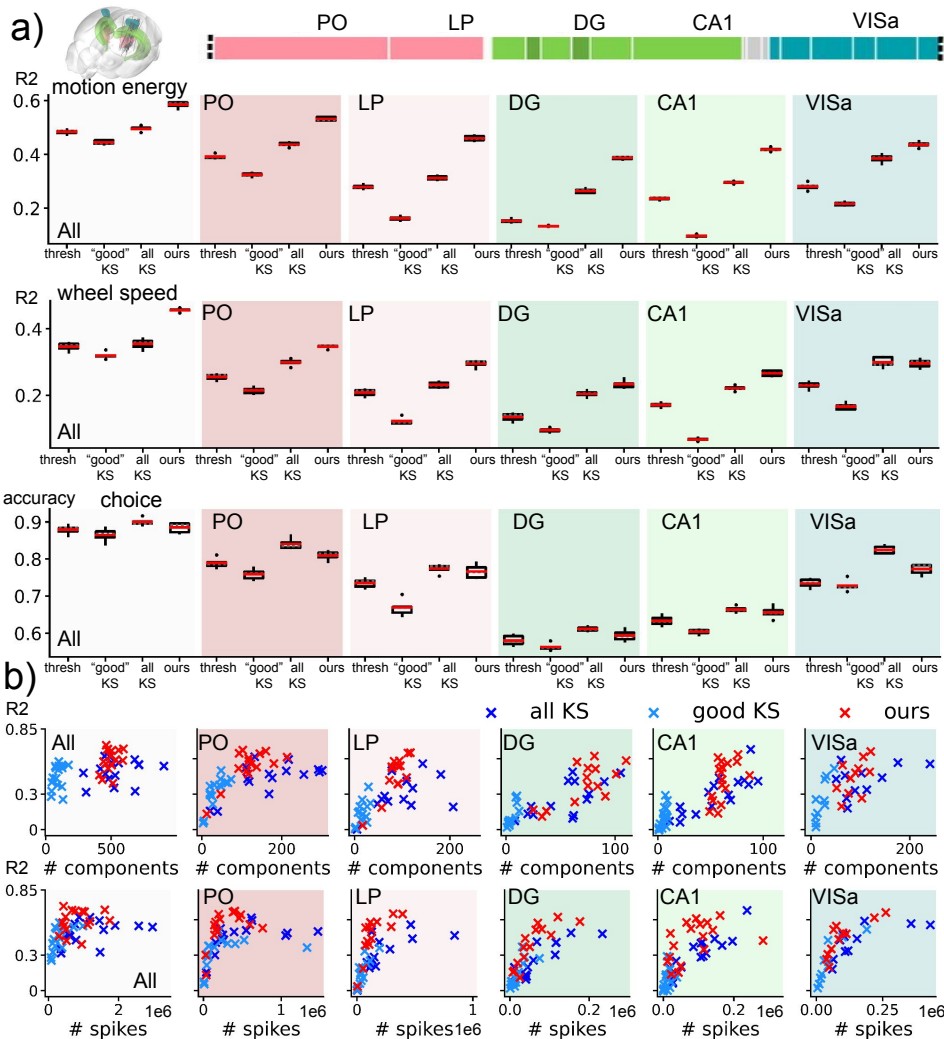

Figure 3: **Decoding comparisons broken down by brain regions.** (a) We decoded 20 IBL datasets acquired using NP1 probes which were inserted into mice performing a behavioral task. The locations of the probe insertions in the mouse brain and the corresponding brain parcellations along the NP1 probe are shown. We compared the performance of all decoders across different recorded brain regions. For the "All" region, spikes from all brain regions were utilized for decoding. In contrast, for the "PO," "LP," "DG," "CA1," and "VISa" regions, only spikes from the respective regions were used for decoding. The decoding performance were summarized using box plots showing metrics obtained through a 5-fold CV and averaged across 20 IBL sessions. We observe a higher accuracy from PO, LP, and VISa regions when decoding choice; decoding results are more comparable across regions for the continuous behavioral variables. Our proposed decoder consistently achieves higher accuracy in decoding the continuous variables. (b) We use scatter plots to quantify the relationship between decoding quality, measured by $R^2$ from decoding motion energy, and the number of components used for decoding. In the case of "all KS" and "good KS", the number of components corresponds to the number of KS units. For our method, the number of components refers to the number of MoG components used. For all methods, the decoding performance is higher when using more components (in the regime of a small number of components). Our decoding method consistently outperforms spike-sorted decoders based on KS 2.5 while tending to need fewer components.

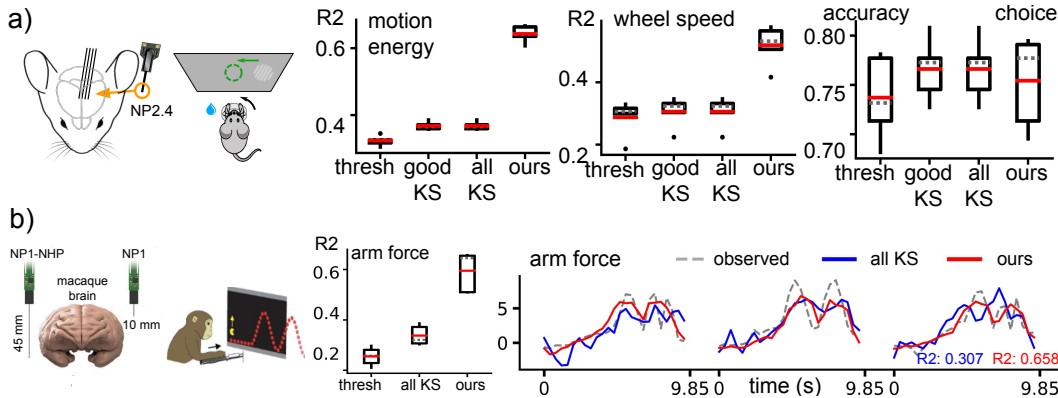

Figure 4: **Decoding performance generalizes across different animals and probe geometry.**
(a) We compare all decoders on a NP2.4 dataset using box plots showing performance metrics
obtained from a 5-fold CV. Our method achieves much higher performance than all other decoders
on continuous behavior decoding with slightly worse choice decoding than the spike-sorted decoder.
(b) We utilize data from a single NP1-NHP recording session to decode the reaching force of a
monkey engaged in a path-tracking (pacman) behavioral task. The decoders are evaluated through
both quantitative analysis (box plots) and qualitative examination of the decoded traces. Each trial
within the NP1-NHP recording has a duration of 9.85 seconds. Our method outperforms all other
decoders on predicting the arm force.

**Different brain regions from 20 datasets.** The decoding results across various brain regions for
20 IBL sessions are summarized in Figure 3. Overall, our approach consistently achieves higher
$R^2$ values compared to other competing methods in decoding both motion energy and wheel speed
across the five recorded brain regions. Notably, decoders based on "good" KS units exhibit poor
performance across all recorded brain regions when compared to decoders based on all KS units. This
observation highlights the importance of utilizing all available information for decoding behaviors
rather than solely relying on "good" units based on sorting quality metrics. The scatter plots in
Figure 3 (b) indicate a general trend where decoding quality tends to increase when more components
(i.e., KS units for the spike-sorted decoders and MoG components for our method) are available for
decoding. However, our method outperforms spike-sorted decoders even with a limited number of
components.

**Different probe geometry.** The decoding results for the NP2.4 and NP1-NHP geometries are
illustrated in Figure 4. For NP2.4, our approach significantly outperforms other competing methods
when decoding motion energy and wheel speed, while again, the approaches are more comparable
when decoding the discrete choice variable (our method performs slightly worse than the spike-
sorted decoder). For NP1-NHP, Figure 4 demonstrates that our method achieves better decoding
performance ($R^2 \approx 0.6$) compared to the spike-thresholded ($R^2 \approx 0.2$) and spike-sorted baselines
($R^2 \approx 0.3$). "Good" KS units are not available in this scenario (the IBL quality criteria were not
applied to this primate dataset) and are therefore not included in the results.

**Comparison to a state-of-the-art clusterless decoder.** We compare our method to a state-of-the-
art clusterless point process decoder (Denovellis et al., 2021) in Table 1. Our method has higher
decoding performance than the point process decoder on both HD and simulated tetrode datasets for
all behavior variables. This performance improvement is likely due to the increased flexibility of our
decoder compared to state-space models that make stronger assumptions about the dynamics of the
decoded signals.

**Computation time.** In Figure 5, we provide a computation time comparison relative to real-time.
Our decoding step operates at a sub-real-time pace (0.3 times real-time). The total time after
preprocessing for our method is close to real-time.

|  | Multiple tetrodes (position) | NP1 (wheel speed) | NP1 (motion energy) |
|---|---|---|---|
| Denovellis et al. (2021) | 0.91 ($\pm$ 0.01) | 0.50 ($\pm$ 0.16) | 0.55 ($\pm$ 0.15) |
| Density-Based | **0.97** ($\pm$ 0.03) | **0.63** ($\pm$ 0.12) | **0.63** ($\pm$ 0.14) |

Table 1: **Comparison to a state-of-the-art clusterless decoder.** We evaluated the performance of both methods using 5-fold cross-validation. We reported the mean correlation between the ground-truth behavior and the decoded behavior along with the standard deviation. All tetrode data was simulated. For the HD datasets, we averaged the results across three IBL datasets.

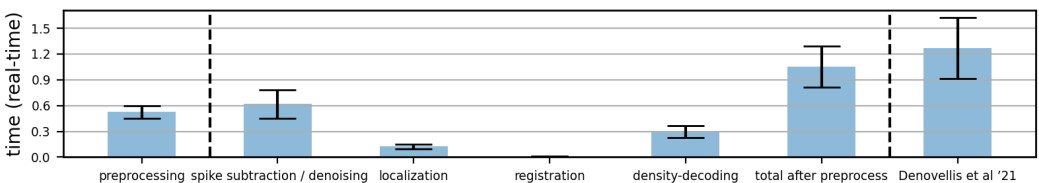

Figure 5: **Computation time measured relative to real-time.** "Preprocessing" includes destriping, required by all decoders (IBL et al., 2022). "Total after preprocess" includes spike subtraction, denoising, localization, registration and density-decoding. The computation time of the clusterless point process decoder (Denovellis et al., 2021) is also provided.

## 5 Discussion

In this work, we introduce a probabilistic model-based neural decoding method that relates spike feature distributions to behavioral correlates for more accurate behavior decoding. Our method is designed for high-density recording devices such as Neuropixels probes, utilizing novel HD spike features (i.e., spike locations) and maximum peak-to-peak amplitudes of the spikes. We further develop an efficient variational approach to perform inference in this model. We benchmark our method across a comprehensive set of HD recordings with varying levels of sorting quality, different probe geometries, and distinct brain regions. We demonstrate that our decoding method can consistently outperform spike-thresholded decoders and spike-sorted decoders across a wide variety of experimental contexts. This motivates a shift towards a spike-feature based decoding paradigm that avoids the need for spike sorting while also achieving comparable or superior decoding performance to approaches relying on well-isolated single-units.

While our method shows promising results, it is essential to explore avenues for improvement. Two potential improvements to our method include utilizing deep learning-based models to capture complex functional associations between firing rates and behaviors and also introducing dependencies among the mixture components to account for correlated neuronal firing patterns. An interesting extension of this work would be to apply this spike-feature based paradigm to unsupervised learning of neural dynamics which would enable us to estimate MoG "firing rates" conditioned on time-varying latent variables. By addressing these challenges and expanding the scope of our method, we can advance our understanding of neural activity and its relationship with behavior in both the supervised and unsupervised settings.

## 6 Broader Impact

Neuroscience research aims to uncover how much behavior information can be decoded from neural signals in specific brain regions. To do this fairly across regions without bias from spike sorting quality, we need decoding methods that do not rely on spike sorting. However, we should recognize the current decoders' limitations to avoid drawing erroneous conclusions from the decoding results. Our generative model, which improves predictive performance by conditioning on external variables, is also relevant to the machine learning community.

## Acknowledgement

This work was supported by grants from the Wellcome Trust (209558 and 216324), National Institutes of Health (1U19NS123716 and K99MH128772) and the Simons Foundation. We thank Matt Whiteway, Tatiana Engel, and Alexandre Pouget for helpful conversations and feedback on the manuscript.

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

## Supplementary Material

## 1 Decoding using automatic differentiation variational inference (ADVI)

In the method section of our paper, we describe the general encoding-decoding paradigm. In this section of the supplementary material, we delve into a specific case that focuses on decoding continuous behaviors, $y_k \in \mathbb{R}^T$, that exhibit temporal variations. We introduce the use of ADVI, which allows us to model the relationship between firing rates of MoG components and behavior correlates using a generalized linear model (GLM).

| | Notation | Definition |
|---|---|---|
| | $s_{itk}$ | spike feature $i$ at time bin $t$ in trial $k$ |
| | $y_k$ | behavior in trial $k$ |
| | $z_{itk}$ | MoG assignment of spike feature $i$ at time bin $t$ in trial $k$ |
| | $\eta_c = (\mu_c, \Sigma_c)$ | mean and covariance matrix of MoG component $c$ |
| | $\pi_{tk}$ | MoG mixing proportion at time bin $t$ in trial $k$ |
| | $n_{tk}$ | number of spikes collected at time bin $t$ in trial $k$ |
| | $C$ | number of MoG components |
| | $T$ | number of time bins within each trial |
| | $K$ | number of trials in a session |
| ADVI | $\lambda_{ctk}$ | behavior-dependent firing rate of component $c$ at time bin $t$ in trial $k$ |
| | $b_c$ | "baseline firing rate" of component $c$ |
| | $\beta_{ct}$ | "behavior-modulated firing rate" of component $c$ at time bin $t$ |
| | $\eta_{b_c} = (\mu_{b_c}, \sigma^2_{b_c})$ | mean and variance of the variational posterior distribution for $b_c$ |
| | $\eta_{\beta_{ct}} = (\mu_{\beta_{ct}}, \sigma^2_{\beta_{ct}})$ | mean and variance of the variational posterior distribution for $\beta_{ct}$ |
| | $\eta_{y_{tk}} = (\mu_{y_{tk}}, \sigma^2_{y_{tk}})$ | mean and variance of the variational posterior distribution for $y_{tk}$ |
| CAVI | $\lambda_{ct1}, \lambda_{ct0}$ | firing rate of component $c$ at time bin $t$ that switches between two states |
| | $\phi$ | variational paramter that represents the probability of $y_k = 1$ |
| | $\rho_{ictk}$ | unnormalized posterior probability of assigning $s_{itk}$ to component $c$ |
| | $r_{ictk}$ | normalized posterior probability of assigning $s_{itk}$ to component $c$ |
| | $y^*_{k1}, y^*_{k0}$ | unnormalized posterior probability of $y_k = 1$ and $y_k = 0$ |
| | $\nu_k$ | normalized posterior probability of $y_k = 1$ |

Table 2: Table of notation.

### 1.1 ADVI-based encoder

Building upon the general model specification outlined in Equations 1-3, we can describe the generative model of the encoder as follows:

$$\lambda_{ctk} = \exp(b_c + \beta_{ct} \cdot y_{tk}), \ b_c \sim \mathcal{N}(b_c; 0, 1), \ \beta_{ct} \sim \mathcal{N}(\beta_{ct}; 0, 1), \tag{6}$$

$$z_{itk} \sim \text{Categorical}(z_{itk}; \pi_{tk}), \ \pi_{tk} = \{\pi_{ctk}\}_{c=1}^C, \ \pi_{ctk} = \frac{\lambda_{ctk}}{\sum_{c'} \lambda_{c'tk}}, \tag{7}$$

$$s_{itk} \sim \mathcal{N}(s_{itk}; \eta_{z_{itk}}), \ \eta_c = (\mu_c, \Sigma_c), \tag{8}$$

where $b$ and $\beta$ are the unknown model parameters corresponding to $\theta$ in Equation 1, and are sampled from a standard normal prior distribution. The latent spike assignment $z$ depends on the mixing proportion $\pi$ of the MoG, which is influenced by the behavior $y$ through the firing rate $\lambda$. Intuitively, we can interpret $\lambda_{ctk}$ as the "firing rate" of component $c$ at time $t$ in trial $k$, while $b_c$ and $\beta_c$ describe the "baseline firing rate" and "behavior-modulated firing rate" of component $c$, respectively.

To learn the latent variables $z$, $b$ and $\beta$, we posit a mean-field Gaussian variational approximation

$$q(z, b, \beta) = \prod_{c,t} q(z_{ct}) q(b_c) q(\beta_{ct}) \tag{9}$$

for the posterior distribution $p(z, b, \beta \mid s, y)$. Obtaining exact updates for $b$ and $\beta$ is challenging due to the normalization term for $\pi$ in Equation 7. Therefore, we employ ADVI to maximize the

ELBO and utilize stochastic gradient ascent to update $b$ and $\beta$. ADVI requires that the model be differentiable with respect to the parameters, and with a normal prior, the latent variables $b$ and $\beta$ reside in the real coordinate space and cause no issues with differentiability. The variational approximations for $b$ and $\beta$ are

$$q(b) = \prod_c q(b_c; \eta_{b_c}) = \prod_c \mathcal{N}(b_c; \eta_{b_c}), \ \eta_{b_c} = (\mu_{b_c}, \sigma_{b_c}^2), \tag{10}$$

$$q(\beta) = \prod_{c,t} q(\beta_{ct}; \eta_{\beta_{ct}}) = \prod_{c,t} \mathcal{N}(\beta_{ct}; \eta_{\beta_{ct}}), \ \eta_{\beta_{ct}} = (\mu_{\beta_{ct}}, \sigma_{\beta_{ct}}^2). \tag{11}$$

A drawback of the parameterization in Equations 6-8 is that the spike assignment variables $z$ are discrete and not compatible with ADVI. An alternative, equivalent parameterization that addresses these problems is to marginalize over $z$. The marginalized model is

$$\lambda_{ctk} = \exp(b_c + \beta_{ct} \cdot y_{tk}), \ b_c \sim \mathcal{N}(b_c; 0, 1), \ \beta_{ct} \sim \mathcal{N}(\beta_{ct}; 0, 1), \tag{12}$$

$$\pi_{ctk} = \frac{\lambda_{ctk}}{\sum_{c'} \lambda_{c'tk}}, \ \pi_{tk} = \{\pi_{ctk}\}_{c=1}^C, \tag{13}$$

$$s_{itk} = \sum_{c=1}^C \pi_{ctk} \mathcal{N}(s_{itk}; \eta_c), \ \eta_c = (\mu_c, \Sigma_c). \tag{14}$$

Under this parameterization, the ELBO for the encoder is

$$\mathcal{L}_{\text{enc}}^{\text{ADVI}} := \mathbb{E}_{q(b,\beta)}[\log p(s, b, \beta \mid y)] - \mathbb{E}_{q(b,\beta)}[\log q(b, \beta)] \tag{15}$$

$$= \mathbb{E}_{q(b,\beta)}\Big[ \sum_{k,t,c,i} \log \mathcal{N}(s_{itk}; \eta_c) + \log \pi_{ctk} + \log \mathcal{N}(b_c; 0, 1) + \log \mathcal{N}(\beta_{ct}; 0, 1) \Big] \tag{16}$$

$$- \mathbb{E}_{q(b,\beta)}\Big[ \sum_{c,t} \log \mathcal{N}(b_c; \eta_{b_c}) + \log \mathcal{N}(\beta_{ct}; \eta_{\beta_{ct}}) \Big].$$

## 1.2   ADVI-based decoder

The decoder adopts the same generative model as the encoder described in Equations 6-8 with two exceptions: 1) The latent variable $y_{tk}$ is assumed to have a standard normal prior, i.e., $y_{tk} \sim \mathcal{N}(y_{tk}; 0, 1)$, assuming independence at each time step $t$. Alternatively, a Gaussian process prior can be chosen to capture temporal correlations between time steps. 2) The parameters $b$ and $\beta$ are learned from the ADVI-based encoder and kept fixed in the decoder.

The mean-field Gaussian variational approximation for the posterior distribution $p(z, y \mid s)$ is

$$q(z, y) = \prod_{c,t} q(z_{ct}) q(y_t), \tag{17}$$

where

$$q(y) = \prod_{k,t} q(y_{tk}; \eta_{y_{tk}}) = \prod_{k,t} \mathcal{N}(y_{tk}; \eta_{y_{tk}}), \ \eta_{y_{tk}} = (\mu_{y_{tk}}, \sigma_{y_{tk}}^2). \tag{18}$$

To enable the use of ADVI, we can marginalize out the discrete latent variable $z$, thereby transforming the MoG model into a differentiable form. Under the marginalized MoG parametrization in Equations 12-14, the ELBO of the decoder is

$$\mathcal{L}_{\text{dec}}^{\text{ADVI}} := \mathbb{E}_{q(y)}[\log p(s, y)] - \mathbb{E}_{q(y)}[\log q(y)] \tag{19}$$

$$= \mathbb{E}_{q(y)}\Big[ \sum_{k,t,c,i} \log \mathcal{N}(s_{itk}; \eta_c) + \log \pi_{ctk} + \log \mathcal{N}(y_{tk}; 0, 1) \Big] \tag{20}$$

$$- \mathbb{E}_{q(y)}\Big[ \sum_{k,t} \log \mathcal{N}(y_{tk}; \eta_{y_{tk}}) \Big].$$

# 2 Decoding using coordinate ascent variational inference (CAVI)

We present a specific scenario for decoding binary variables, $y_k \in \{0, 1\}$, where we derive exact updates for the variational variables using CAVI.

## 2.1 CAVI-based encoder

Extending the general model described in Equations 1-3, the generative model of the encoder can be defined as follows:

$$\pi_{ctk} = \left( \frac{\lambda_{ct1}}{\sum_{c'} \lambda_{c't1}} \right)^{y_k} \left( \frac{\lambda_{ct0}}{\sum_{c'} \lambda_{c't0}} \right)^{1-y_k}, \tag{21}$$

$$z_{itk} \sim \text{Categorical}(z_{itk}; \pi_{tk}), \ \pi_{tk} = \{\pi_{ctk}\}_{c=1}^{C}, \tag{22}$$

$$s_{itk} \sim \mathcal{N}(s_{itk}; \eta_{z_{itk}}), \ \eta_c = (\mu_c, \Sigma_c), \tag{23}$$

where $z$ and $\lambda$ are the latent variables that we aim to learn. The behavior-dependent firing rates of each component $c$ at time $t$ vary based on the binary variable $y$, such that the components switch between two behavioral states characterized by firing rates $\lambda_{ct1}$ and $\lambda_{ct0}$.

The log-likelihood of the encoder can be written as

$$\log p(s, z \mid y) = \sum_{k,t,c,i} z_{ictk} \big\{ \log \mathcal{N}(s_{itk}; \eta_c) + y_k (\log \lambda_{ct1} - \log \Lambda_{t1}) \tag{24}$$
$$+ (1 - y_k)(\log \lambda_{ct0} - \log \Lambda_{t0}) \big\},$$

where $\Lambda_{t1} = \sum_{c'} \lambda_{c't1}$ and $\Lambda_{t0} = \sum_{c'} \lambda_{c't0}$. To approximate the posterior $p(z \mid s, y)$, we employ the mean-field variational approximation $q(z) = \prod_{c,t} q(z_{ct})$. The ELBO of the encoder is

$$\mathcal{L}_{\text{enc}}^{\text{CAVI}} := \mathbb{E}_{q(z)}[\log p(s, z \mid y)] - \mathbb{E}_{q(z)}[\log q(z)]. \tag{25}$$

The exact update for $q(z)$ is obtained by maximizing the ELBO with respect to $q(z)$, which leads to the following update equation:

$$q(z_{ct}) \propto \exp\{\mathbb{E}_{q_{-ct}}[\log p(z_{ct}, z_{-ct}, s \mid y)]\}, \tag{26}$$

where $q_{-ct}$ means $\prod_{c' \neq c} \prod_{t' \neq t} q(z_{c't'})$. Then,

$$q(z_{ictk} = 1) \propto \exp \left\{ \log \mathcal{N}(s_{itk}; \eta_c) + y_k \log \left( \frac{\lambda_{ct1}}{\Lambda_{t1}} \right) + (1 - y_k) \log \left( \frac{\lambda_{ct0}}{\Lambda_{t0}} \right) \right\} := \rho_{ictk} \tag{27}$$

denotes the unnormalized posterior probability of assigning spike $i$ collected at time $t$ in trial $k$ to component $c$, while $\mathbb{E}[z_{ictk}] = \rho_{ictk} / \sum_{c'} \rho_{ic'tk} := r_{ictk}$ represents the normalized posterior probability.

After fixing $q(z)$, the term in the ELBO which depends on $\lambda$, $\mu$ and $\Sigma$ can be expressed as

$$\mathcal{L} := \sum_{k,t,c,i} r_{ictk} \big\{ \log \mathcal{N}(s_{itk}; \eta_c) + y_k \log \left( \frac{\lambda_{ct1}}{\Lambda_{t1}} \right) + (1 - y_k) \log \left( \frac{\lambda_{ct0}}{\Lambda_{t0}} \right) \big\}.$$

**Algorithm 1** CAVI-based encoder

---

**Input:** $\{s_{itk}\}, \{y_k\}, i = 1 : n_{tk}, t = 1 : T, k = 1 : K$, number of components $C$.
  Initialize $\{\mu_c, \Sigma_c\}, c = 1 : C$.
  **while** ELBO not converged **do**
    **for all** $k \in 1 : K$ **do**
      **for all** $t \in 1 : T$ **do**
        **for all** $i \in 1 : n_{tk}$ **do**
          Set $q(z_{ictk} = 1) \propto \rho_{ictk}$.                    ▷ eq. (27)
        **end for**
      **end for**
    **end for**

    **for all** $c \in 1 : C$ **do**
      Set $\mu_c = \mu_c^*, \Sigma_c = \Sigma_c^*$.                          ▷ eq. (36-39)
      **for all** $t \in 1 : T$ **do**
        Set $\lambda_{ct0} = \lambda_{ct0}^*, \lambda_{ct1} = \lambda_{ct1}^*$.             ▷ eq. (32-33)
      **end for**
    **end for**
    Compute the ELBO $\mathcal{L}_{\text{enc}}^{\text{CAVI}}$.                       ▷ eq. (25)
  **end while**
  Return $q(z), \lambda, \mu, \Sigma$.

---

We derive the update for $\lambda$ by setting the gradients $\nabla_{\lambda_{ct1}} \mathcal{L}$ and $\nabla_{\lambda_{ct0}} \mathcal{L}$ to 0:

$$\nabla_{\lambda_{ct1}} \mathcal{L} = \nabla_{\lambda_{ct1}} \sum_{k,i} r_{ictk} \cdot y_k (\log \lambda_{ct1} - \log \Lambda_{t1}) \tag{28}$$

$$= \frac{\sum_{k,i} r_{ictk} \cdot y_k}{\lambda_{ct1}} - \frac{\sum_{k,i} \sum_{c'=1}^{C} r_{ic'tk} \cdot y_k}{\Lambda_{t1}} = 0 \tag{29}$$

$$\implies \sum_{k,i} r_{ictk} \cdot y_k (\lambda_{ct1} + \sum_{c' \neq c} \lambda_{c't1}) = \sum_{k,i} \sum_{c'=1}^{C} r_{ic'tk} \cdot y_k \cdot \lambda_{ct1} \tag{30}$$

$$\implies \sum_{k,i} y_k (\sum_{c' \neq c} r_{ic'tk}) \lambda_{ct1} = \sum_{k,i} y_k \cdot r_{ictk} (\sum_{c' \neq c} \lambda_{c't1}) \tag{31}$$

$$\implies \lambda_{ct1} = \frac{\sum_{k,i} y_k \cdot r_{ictk} (\sum_{c' \neq c} \lambda_{c't1})}{\sum_{k,i} y_k (\sum_{c' \neq c} r_{ic'tk})} := \lambda_{ct1}^*. \tag{32}$$

$$\nabla_{\lambda_{ct0}} \mathcal{L} = 0 \implies \lambda_{ct0} = \frac{\sum_{k,i} (1 - y_k) r_{ictk} (\sum_{c' \neq c} \lambda_{c't0})}{\sum_{k,i} (1 - y_k) (\sum_{c' \neq c} r_{ic'tk})} := \lambda_{ct0}^*. \tag{33}$$

Consider the gradient with respect to the $\eta_c$ parameter,

$$\nabla_{\eta_c} \mathcal{L} = \sum_{k,t,i} r_{ictk} \nabla_{\eta_c} \log \mathcal{N}(s_{itk}; \eta_c) \tag{34}$$

$$= \sum_{k,t,i} r_{ictk} \nabla_{\eta_c} \frac{1}{2} \left( \log |\Sigma_c^{-1}| - \text{Tr}\{\Sigma_c^{-1} (s_{itk} - \mu_c)(s_{itk} - \mu_c)^\top\} \right). \tag{35}$$

The closed-form updates for $\mu_c$ and $\Sigma_c$ are

$$\nabla_{\mu_c}\mathcal{L} = \sum_{k,t,i} r_{ictk}\Sigma_c^{-1}(s_{itk} - \mu_c) = 0 \tag{36}$$

$$\implies \mu_c = \frac{1}{n_c}\sum_{k,t,i} r_{ictk}s_{itk} := \mu_c^*, \quad n_c = \sum_{k,t,i} r_{ictk}. \tag{37}$$

$$\nabla_{\Sigma_c}\mathcal{L} = \frac{1}{2}\sum_{k,t,i} r_{ictk}(\Sigma_c - (s_{itk} - \mu_c)(s_{itk} - \mu_c)^\top) = 0 \tag{38}$$

$$\implies \Sigma_c = \frac{1}{n_c}\sum_{k,t,i} r_{ictk}(s_{itk} - \mu_c)(s_{itk} - \mu_c)^\top := \Sigma_c^*. \tag{39}$$

## 2.2 CAVI-based decoder

The CAVI-based decoder employs the same generative model as the CAVI-based encoder, with the exception that the behavior-dependent firing rates $\lambda_{ct1}$ and $\lambda_{ct0}$ are learned by the encoder and kept fixed, and the behavior $y$ is treated as an unknown latent variable. We sample $y$ from a prior distribution, $y_k \sim \text{Bernoulli}(\phi)$, where $\phi$ is a variational parameter that represents the probability that $y_k = 1$. The log-likelihood can be expressed as follows

$$\log p(s, z, y) = \sum_{k,t,c,i} z_{ictk}\{\log\mathcal{N}(s_{itk}; \eta_c) + y_k(\log\lambda_{ct1} - \log\Lambda_{t1}) \tag{40}$$

$$+ (1 - y_k)(\log\lambda_{ct0} - \log\Lambda_{t0})\} + y_k\log\phi + (1 - y_k)\log(1 - \phi).$$

We use the factorization $q(z, y) = q(z)q(y) = \prod_{c,t} q(z_{ct})q(y)$ to approximate the posterior distribution $p(z, y \mid s)$. The ELBO of the decoder can be defined as

$$\mathcal{L}_{\text{dec}}^{\text{CAVI}} := \mathbb{E}_{q(z,y)}[\log p(s, z, y)] - \mathbb{E}_{q(z,y)}[\log q(z, y)]. \tag{41}$$

---

**Algorithm 2** CAVI-based decoder

---

**Input:** $\{s_{itk}\}, \{\lambda_{ct0}, \lambda_{ct1}\}, i = 1 : n_{tk}, t = 1 : T, k = 1 : K, c = 1 : C$.
  Initialize $\{\mu_c\}, \{\Sigma_c\}$.
  **while** ELBO not converged **do**
    **for all** $k \in 1 : K$ **do**
      Set $q(y_k = 1) \propto y_{k1}^*$.                                         ▷ eq. (44)
      **for all** $t \in 1 : T$ **do**
        **for all** $i \in 1 : n_{tk}$ **do**
          Set $q(z_{ictk} = 1) \propto \rho_{ictk}$.                          ▷ eq. (43)
        **end for**
      **end for**
    **end for**

    **for all** $c \in 1 : C$ **do**
      Set $\mu_c = \mu_c^*, \Sigma_c = \Sigma_c^*$.                                ▷ eq. (36-39)
    **end for**
    Set $\phi = \phi^*$.                                               ▷ eq. (46)
    Compute the ELBO $\mathcal{L}_{\text{dec}}^{\text{CAVI}}$.                        ▷ eq. (41)
  **end while**
  Return $q(z, y), \phi, \mu, \Sigma$.

---

The exact updates for $q(z)$ and $q(y)$ that guarantee an increase in the ELBO are

$$q(z) \propto \exp\{\mathbb{E}_{q(y)}[\log p(s, z, y)]\}, \quad q(y) \propto \exp\{\mathbb{E}_{q(z)}[\log p(s, z, y)]\}, \tag{42}$$

where

$$q(z_{ictk} = 1) \propto \exp\{\log\mathcal{N}(s_{itk}; \eta_c) + \mathbb{E}[y_k]\log\left(\tfrac{\lambda_{ct1}}{\Lambda_{t1}}\right) + (1 - \mathbb{E}[y_k])\log\left(\tfrac{\lambda_{ct0}}{\Lambda_{t0}}\right)\} := \rho_{ictk}, \tag{43}$$

and $\mathbb{E}[z_{ictk}] = \rho_{ictk}/\sum_{c'}\rho_{ic'tk} := r_{ictk}$ are the unnormalized and normalized posterior probabilities of assigning spike $i$ collected at time $t$ in trial $k$ to component $c$, respectively. The unnormalized posterior probabilities of $y_k = 1$ and $y_k = 0$ are

$$q(y_k = 1) \propto \exp\{\sum_{t,c,i} \mathbb{E}[z_{ictk}](\log\lambda_{ct1} - \log\Lambda_{t1}) + \log\phi\} := y_{k1}^*, \tag{44}$$

$$q(y_k = 0) \propto \exp\{\sum_{t,c,i} \mathbb{E}[z_{ictk}](\log\lambda_{ct0} - \log\Lambda_{t0}) + \log(1 - \phi)\} := y_{k0}^*,$$

and $\mathbb{E}[y_k] = y_{k1}^*/(y_{k1}^* + y_{k0}^*) := \nu_k$ represents the normalized posterior probability of $y_k = 1$.

After fixing $q(z)$ and $q(y)$, the term in the ELBO which depends on $\phi$, $\mu$ and $\Sigma$ can be written as

$$\mathscr{L}' := \sum_{k,t,c,i} r_{ictk}\{\log\mathcal{N}(s_{itk}; \eta_c) + \nu_k(\log\lambda_{ct1} - \log\Lambda_{t1}) \tag{45}$$
$$+ (1 - \nu_k)(\log\lambda_{ct0} - \log\Lambda_{t0})\} + \nu_k\log\phi + (1 - \nu_k)\log(1 - \phi).$$

Considering the gradient of the ELBO with respect to the $\phi$ parameter, we obtain its update:

$$\nabla_\phi\mathscr{L}' = 0 \implies \phi^* = \frac{1}{K}\sum_{k=1}^{K}\nu_k. \tag{46}$$

The updates for $\eta_c$ are computed in a similar manner as described in Equations 36-39.

## 3    MoG initialization

We employ the following procedure to intialize the MoG model used in both the ADVI-based and CAVI-based algorithms: 1) According to Figure 1 (a), the spike feature distribution is highly multimodal. To determine the appropriate number of modes, we utilize *isosplit* (Magland and Barnett, 2015) to cluster the spike features. This step helps in splitting the set of spike features into distinct clusters. 2) For each identified cluster, we compute the mean and variance of the spike features belonging to that cluster, which serve as the parameters for the corresponding Gaussian component. This automatic selection of the number of MoG components and the initialization of means and covariance matrices facilitate the initialization of the ADVI-based and CAVI-based algorithms.

# 4 Data preprocessing

We provide a brief overview of our data preprocessing pipeline, which involves the following steps.

**Destriping.** During the data collection process, we encounter line noise due to voltage leakage on the probe. This translates into large "stripes" of noise spanning the whole probe. To mitigate the impact of these noise artifacts, we apply a destriping procedure (Chapuis et al., 2022).

**Subtraction-based spike detection and denoising.** We employ the iterative subtraction-based procedure for spike detection and collision-correction described in Boussard et al. (2023).

**Spike localization.** We employ the method of Boussard et al. (2021) to estimate the location of each denoised spike.

**Drift registration.** Probe motion (or drift) in the electrophysiology data poses a challenge for downstream analyses. Decentralized registration (Windolf et al., 2022) is applied to track and correct for motion drift in the high-density probe recordings.

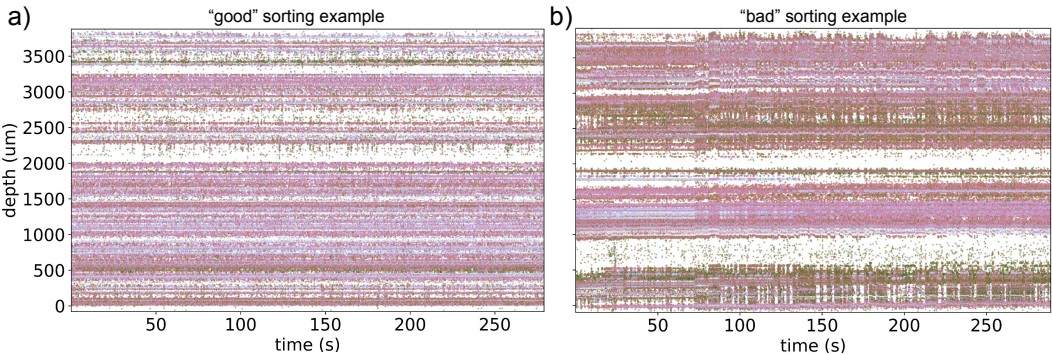

Figure 6: **Motion drift in "good" and "bad" sorting recordings.** (a) The motion-registered spike raster plot of a "good" sorting example that is less affected by drift. (b) The spike raster plot of a "bad" sorting example, which is still affected by drift even after registration.

# 5 Behavior decoder

To decode binary behaviors, such as the mouse's left or right choices, we utilize $\mathcal{L}_2$-penalized logistic regression. For decoding dynamic behaviors, such as wheel speed, we employ a sliding-window algorithm to aggregate the entries of the weight matrix, $W_{kct}$, over time. Within the time window $[t - \delta, t + \delta]$, where $\delta$ is the window size, we stack $2\delta$ weight matrix entries, $\{W_{kct}\}_{c=1}^{C}$, for time point $t$ in trial $k$. This aggregated weight matrix is then used as input for ridge regression to predict the behavior $y_{tk}$ at time $t$. The window size $\delta$ and the regularization strength are model hyper-parameters, set through cross-validation to achieve the optimal decoding performance.

# 6 Model interpretation

In this section, we provide visualizations to gain insights into the effectiveness of our proposed decoder. We quantify the posterior entropy of each spike assignment in Figure 7 (a). Spike assignments with higher entropy correspond to a spread of posterior probabilities among multiple MoG components. In contrast, traditional spike sorting or thresholding methods result in deterministic spike assignments, leading to lower entropy and empirically reduced decoding performance. In Figure 7 (b), we compare the trial-averaged weight matrices ($W$) used for decoding between spike-sorted, spike-thresholded, and our proposed decoders.

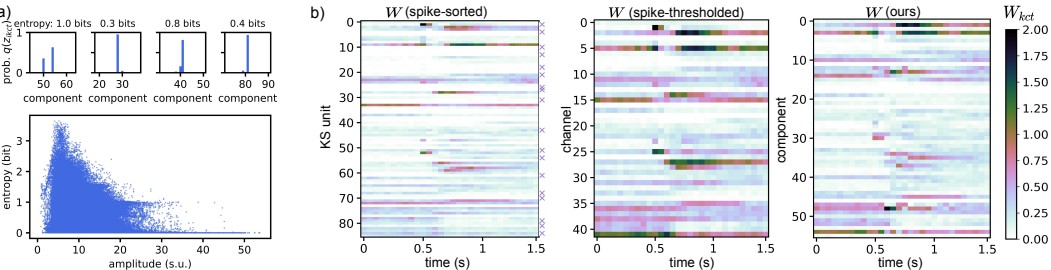

Figure 7: **Model interpretation.** (a) The posterior entropy of the spike assignment is high, when the posterior probability of spike assignment, $q(z_{ikct})$, is spread out among several MoG components instead of being concentrated at a single component. The scatter plot shows that low-amplitude spikes that are difficult to assign have higher posterior entropy than high-amplitude spikes. (b) Visualizations of the averaged weight matrices $W$'s across trials in an example IBL session. For "$W$ (spike-sorted and spike-thresholded)", the $W$ matrix has one-hot rows and each entry $W_{kct}$ is the spike count that belongs to KS unit (channel) $c$ and time bin $t$ in trial $k$. The purple crosses indicate the "good" KS units. For "$W$ (ours)", $W_{kct}$ is the sum of posterior probabilities of spike assignments, and the MoG mixing proportion $\pi$ depends on the behavior $y$ and changes over time. The arrangement of KS units (channels or MoG components) on the heat maps is based on the depth of the NP probe, ensuring comparability across the displayed $W$ matrices.

# 7 Decoding across brain regions

In addition to the previously mentioned five brain regions (PO, LP, DG, CA1, VISa) depicted in Figure 3, we expanded our analysis to include two additional brain regions situated in the cerebellum: the arbor vitae (ARB) and the ansiform cruciform lobule (ANCR). We specifically include the cerebellum in our analysis due to the frequent occurrence of spike sorting issues in this area. In Figure 8 and 9, we present a comparison between our decoder and the spike-sorted decoder that utilizes all KS units across all the brain regions studied. According to Figure 8, our method consistently outperforms the decoder that relies on all KS units across all brain regions and the majority of IBL sessions. Furthermore, Figure 9 specifically demonstrates that our method consistently achieves superior decoding performance in the recorded regions of the cerebellum, where spike sorting quality issues are commonly encountered. This highlights the robustness and reliability of our method, particularly in challenging recording conditions.

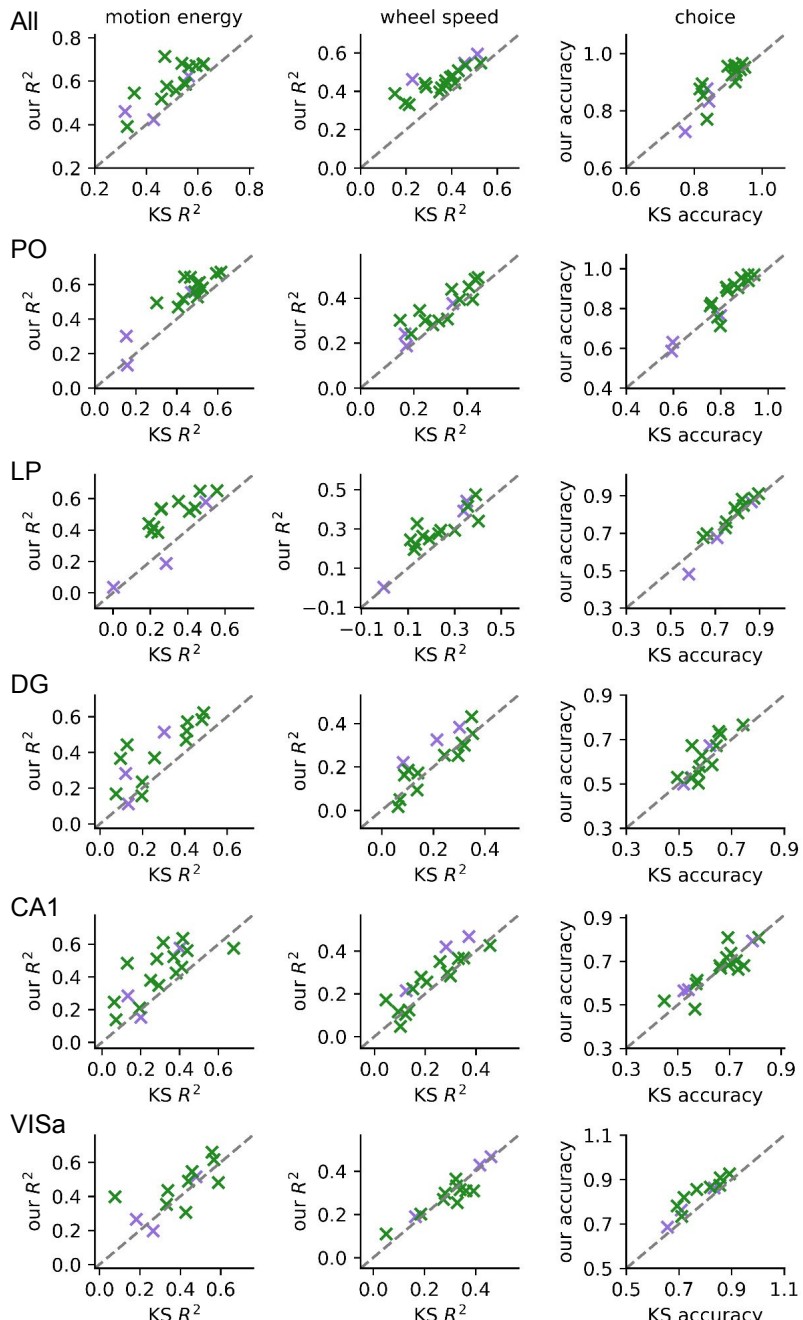

Figure 8: **Decoding comparisons across brain regions in the thalamus, hippocampus and visual cortex.** We compare our decoder to the spike-sorted decoder using all KS units across various brain regions and behavioral tasks. Each point in the scatter plot represents one session from the 20 IBL session previously described in the experiments section. Sessions with "good" sorting quality are depicted in green, while sessions with "bad" sorting quality are shown in purple. The majority of the sessions lie above the gray diagonal line, indicating that our method consistently outperforms the decoder relying on all KS units.

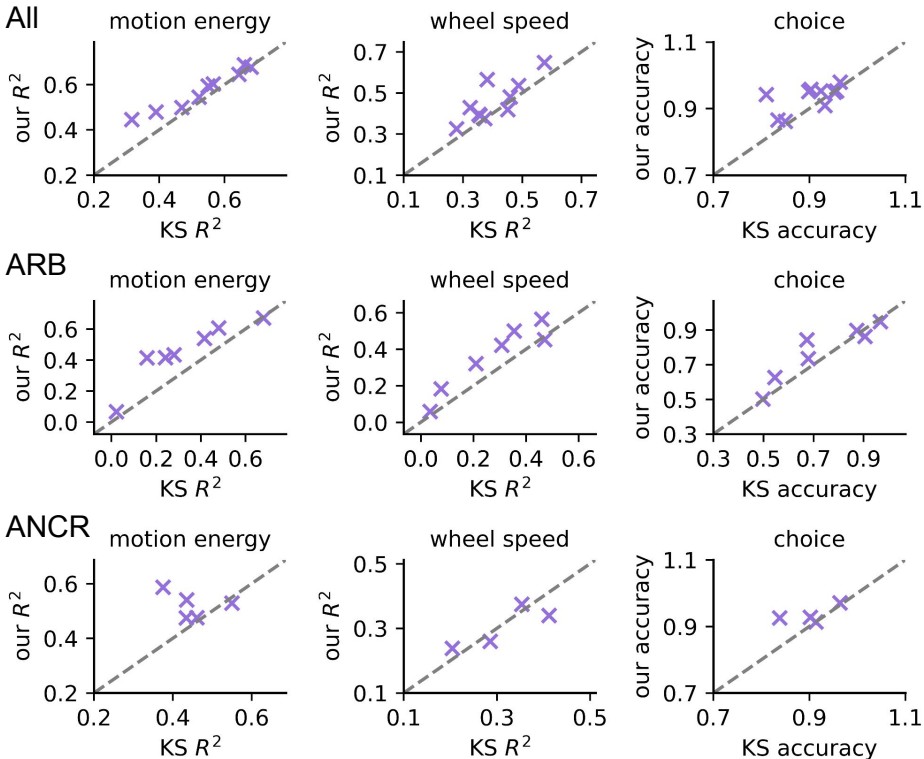

Figure 9: **Decoding comparisons across brain regions in the cerebellum.** We evaluate our decoder against the spike-sorted decoder utilizing all KS units across various brain regions in the cerebellum. The scatter plot visualizes the results for each IBL session in the study, with all sessions depicted in purple indicating "bad" sorting quality. Note that the probes used in the cerebellum sessions were not uniformly implanted in the same set of brain regions. As a result, the number of available sessions for decoding in certain cerebellum regions is limited. Notably, the majority of sessions are positioned above the gray diagonal line, indicating that our method consistently achieves better decoding performance compared to the decoder relying on all KS units. This outcome highlights the robustness of our method, particularly in the context of the cerebellum where spike sorting quality issues are prevalent.

# 8 Ablation study

We conduct an ablation study to investigate the importance of various components in our decoding paradigm. Specifically, we examine how the integration of dynamic (behavior-dependent) mixing proportions in the MoG and higher-dimensional spike features influence decoding performance. Additionally, we analyze how the inclusion criteria for spike waveforms can affect decoding outcomes.

**Effects of dynamic MoG mixing proportion.**    Table 3 presents a comparison between the ordinary MoG with a fixed mixing proportion (referred to as "fixed $\pi$") and our proposed model with a dynamic mixing proportion (referred to as "dynamic $\pi$"). The results indicate that this approach leads to improved decoding performance compared to using a fixed mixing proportion in the MoG model.

| | Motion energy ($R^2$) | | Wheel speed ($R^2$) | | Choice (accuracy) | |
|---|---|---|---|---|---|---|
| | Fixed $\pi$ | Dynamic $\pi$ | Fixed $\pi$ | Dynamic $\pi$ | Fixed $\pi$ | Dynamic $\pi$ |
| All | 0.664 ($\pm$ 0.034) | **0.742** ($\pm$ 0.028) | 0.470 ($\pm$ 0.062) | **0.564** ($\pm$ 0.045) | 0.948 ($\pm$ 0.038) | **0.957** ($\pm$ 0.036) |
| PO | 0.365 ($\pm$ 0.031) | **0.488** ($\pm$ 0.046) | 0.520 ($\pm$ 0.019) | **0.670** ($\pm$ 0.015) | 0.844 ($\pm$ 0.016) | **0.861** ($\pm$ 0.035) |
| LP | 0.145 ($\pm$ 0.015) | **0.464** ($\pm$ 0.054) | 0.114 ($\pm$ 0.032) | **0.342** ($\pm$ 0.027) | 0.917 ($\pm$ 0.026) | **0.931** ($\pm$ 0.022) |
| DG | 0.280 ($\pm$ 0.033) | **0.492** ($\pm$ 0.042) | 0.221 ($\pm$ 0.040) | **0.381** ($\pm$ 0.035) | 0.669 ($\pm$ 0.084) | **0.722** ($\pm$ 0.050) |
| CA1 | 0.407 ($\pm$ 0.021) | **0.538** ($\pm$ 0.038) | 0.308 ($\pm$ 0.051) | **0.428** ($\pm$ 0.030) | 0.621 ($\pm$ 0.062) | **0.626** ($\pm$ 0.064) |
| VISa | 0.488 ($\pm$ 0.046) | **0.490** ($\pm$ 0.047) | 0.318 ($\pm$ 0.056) | **0.364** ($\pm$ 0.074) | 0.857 ($\pm$ 0.065) | **0.874** ($\pm$ 0.066) |

Table 3: Effects of dynamic mixing proportion of the MoG on decoding performance.

**Effects of higher-dimensional spike features.**    In Table 4, we provide a comparison of decoding performance using two different sets of spike features. The first set includes spike location along the width and depth dimensions of the probe (denoted as $x$ and $z$) as well as the maximum peak-to-peak amplitude of the spike (denoted as $a$). The second set includes the first and second principal components (PCs) of the spike waveforms (denoted as $u_1$ and $u_2$) in addition to $x, z$ and $a$. The spike features are visually represented using scatter plots in Figure 10. We report the mean and standard deviation of the decoding accuracy ($R^2$) obtained from a 5-fold CV on a single session. Table 4 provides insights regarding the inclusion of additional waveform PC features for decoding. The findings suggest that the incorporation of these additional PC features does not contribute to significant improvements in decoding performance.

| | Motion energy ($R^2$) | | Wheel speed ($R^2$) | | Choice (accuracy) | |
|---|---|---|---|---|---|---|
| | $(x, z, a)$ | $(x, z, a, u_1, u_2)$ | $(x, z, a)$ | $(x, z, a, u_1, u_2)$ | $(x, z, a)$ | $(x, z, a, u_1, u_2)$ |
| All | **0.531** ($\pm$ 0.026) | 0.529 ($\pm$ 0.026) | **0.484** ($\pm$ 0.042) | 0.478 ($\pm$ 0.045) | 0.917 ($\pm$ 0.019) | 0.917 ($\pm$ 0.019) |
| PO | **0.462** ($\pm$ 0.038) | 0.457 ($\pm$ 0.039) | **0.469** ($\pm$ 0.061) | 0.464 ($\pm$ 0.051) | 0.853 ($\pm$ 0.019) | 0.853 ($\pm$ 0.025) |
| LP | **0.490** ($\pm$ 0.026) | 0.489 ($\pm$ 0.029) | **0.479** ($\pm$ 0.035) | 0.473 ($\pm$ 0.013) | **0.864** ($\pm$ 0.022) | 0.849 ($\pm$ 0.036) |
| DG | **0.335** ($\pm$ 0.028) | 0.321 ($\pm$ 0.037) | **0.273** ($\pm$ 0.030) | 0.264 ($\pm$ 0.025) | 0.675 ($\pm$ 0.038) | **0.679** ($\pm$ 0.049) |
| CA1 | **0.449** ($\pm$ 0.027) | 0.440 ($\pm$ 0.046) | **0.329** ($\pm$ 0.044) | 0.328 ($\pm$ 0.040) | 0.755 ($\pm$ 0.045) | **0.758** ($\pm$ 0.053) |
| VISa | **0.270** ($\pm$ 0.021) | 0.237 ($\pm$ 0.017) | **0.225** ($\pm$ 0.024) | 0.206 ($\pm$ 0.013) | 0.725 ($\pm$ 0.048) | **0.732** ($\pm$ 0.054) |

Table 4: Effects of incorporating higher-dimensional spike features on decoding performance.

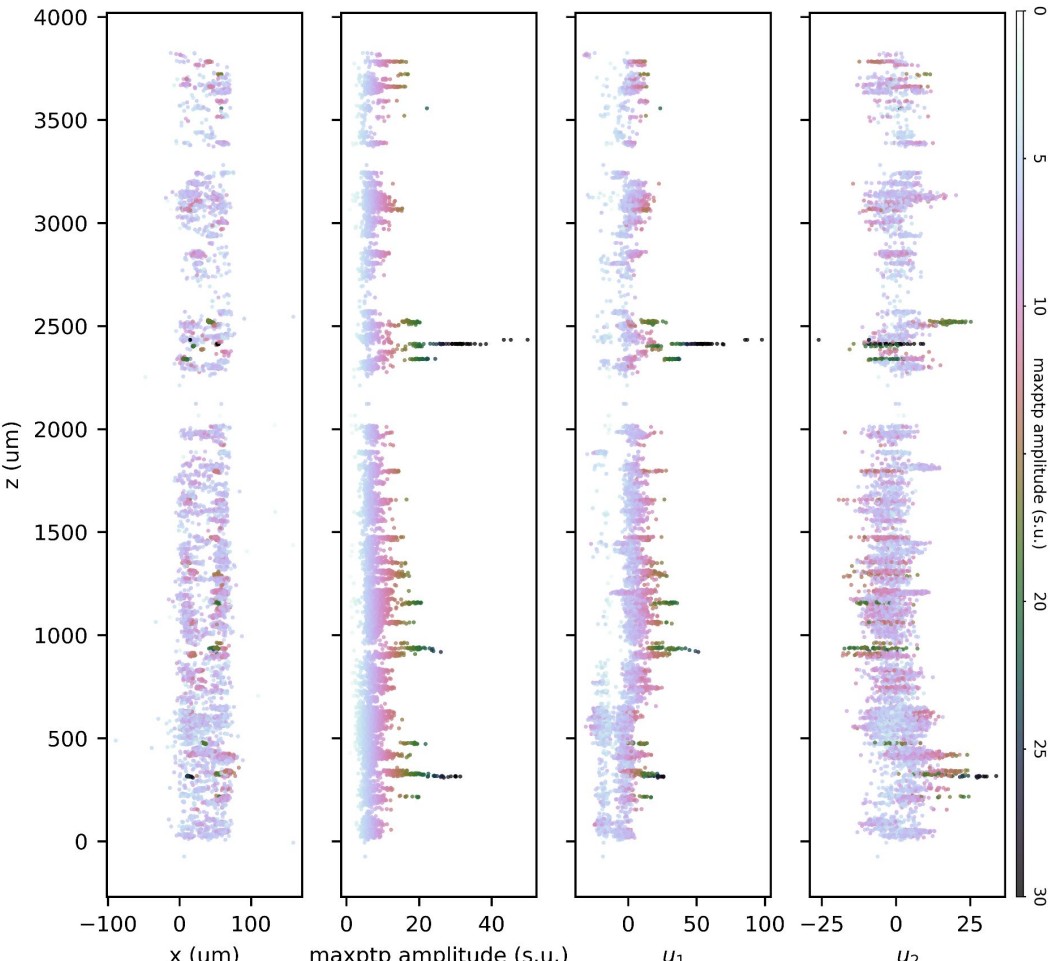

Figure 10: **Visualizations of spike features employed for decoding.** Spike localization features, $(x, z)$, the locations of spikes along the width and depth of the NP1 probe, and waveform features, $a$, the maximum peak-to-peak (max ptp) amplitudes of spikes. Amplitude is measured in standard units (s.u.). $u_1$ and $u_2$ denote the first and second principal components (PCs) of the spike waveforms.

**Effects of inclusion criteria for spike waveforms.** To investigate whether the density-based decoder is performing better by using additional spikes that a spike sorter would miss, we conducted an experiment using different inclusion criteria for spike waveforms. We fitted our model using only spikes detected by Kilosort 2.5, and compared its performance to decoders using spike-sorted outputs and our subtraction-based spike detection on choice and motion energy decoding. The results are summarized in Table 5. As shown in the table, our decoder can achieve comparable or better decoding performance than the spike-sorted decoder when modeling the same spikes. This suggests that the gain in decoding performance can be attributed to the density-based approach, instead of the spike inclusion criteria.

|                                    | Choice (accuracy)    | Motion energy ($R^2$)   |
| ---------------------------------- | -------------------- | ----------------------- |
| Density-based (subtraction spikes) | 0.876 ($\pm$ 0.068)  | **0.589** ($\pm$ 0.111) |
| Density-based (KS spikes)          | 0.876 ($\pm$ 0.079)  | 0.579 ($\pm$ 0.121)     |
| Sorted (KS spikes)                 | **0.887** ($\pm$ 0.078) | 0.503 ($\pm$ 0.117)  |

Table 5: Effects of inclusion criteria for spike waveforms on decoding performance.

# 9 Comparison to a state-of-the-art clusterless decoder

In this section, we outline the specific experimental setup for evaluating our density-based decoder in comparison with the clusterless point process decoder (Denovellis et al., 2021) on both multiple tetrodes and high-density (HD) probes data.

**Application to tetrodes.** We utilized the code provided in the GitHub repository[1] of the clusterless point process decoder (Denovellis et al., 2021) to generate simulated neural and behavioral data. This synthetic dataset was designed to mimic recordings from 5 tetrodes, with each tetrode containing 4 channels. Spike amplitudes from each channel of these multiple tetrodes were selected as the spike features for both decoding methods. The objective of the decoding was to estimate the animal's position from these simulated spike features. As the original simulated position was too straightforward to decode, we intentionally distorted it by blending it with real position data sourced from the GitHub repository[2]. Moreover, we introduced random Gaussian noise to the simulated position for added complexity.

**Application to HD probes.** We evaluated both decoders on decoding wheel speed and motion energy from spike features extracted from NP1 probes across three IBL datasets. To preprocess the data, we followed the pipeline outlined above, extracting spike localization features and maximum peak-to-peak amplitudes as common features for both decoders.

We only focus on continuous behaviors for decoding since the clusterless point process decoder is designed exclusively for continuous behaviors. Notably, due to its continuous time nature, the clusterless point process decoder directly decoded behaviors without time binning. In contrast, our density-based model required time binning of both behavioral and spike feature data into equal-sized time intervals. Consequently, we used the time-binned behaviors for decoding with the density-based approach. We employed 5-fold cross-validation to assess the decoding performance of both decoders. We used a random walk as the state transition for the clusterless point process decoder, and used the estimated variance from the animal's behavior to set the variance of the random walk. The clusterless point process decoder uses a grid-based approximation of the inferred behavior, discretizing the behavior space into place bins; see Section "Decoding" in Denovellis et al. (2021). We determine place bin size based on the square root of the observed behavior variance. Denovellis et al. (2021) use kernel density estimation (KDE) to estimate the distributions of both the behavior variable and the spike features used for decoding. We set the KDE bandwidth that determines the amount of smoothing done for spike features to be 1.0, and the bandwidth for behavior to be the square root of the observed behavior variance; see Section "Encoding - clusterless" in Denovellis et al. (2021).

---

[1]https://github.com/Eden-Kramer-Lab/replay_trajectory_classification
[2]https://github.com/nelpy/example-analyses/blob/master/LinearTrackDemo.ipynb

# 10 Simulation for model validation

We conducted simulations to illustrate the principles of our method. The simulation aimed to show that our encoding model can learn the relationship between spike features and behaviors. We performed two tasks, decoding a binary variable, $y_k$, simulated from a Bernoulli distribution, and decoding a continuous variable, $y_k$, simulated from a Gaussian process. To mimic the data-generating process, we selected Gaussian components with "templates" extracted from a real dataset. The encoding model parameters, $b$ and $\beta$, were also taken from learned parameters in the same dataset. Given $b$, $\beta$ and $y_k$, we simulated the "firing rates" $\lambda$ for each Gaussian component in the mixture, as described in the Method section of our paper. Next, we generated spike features based on these simulated "firing rates," and applied the encoding model to infer the behavior-dependent $\lambda$. Figure 11 displays the learned $\lambda$ for each component $c$, time $t$, and trial $k$. The learned "firing rates" closely resembled the simulated ones, indicating the model's ability to recover the primary associations between spike features and behaviors. With such associations, the decoding model can decode behaviors.

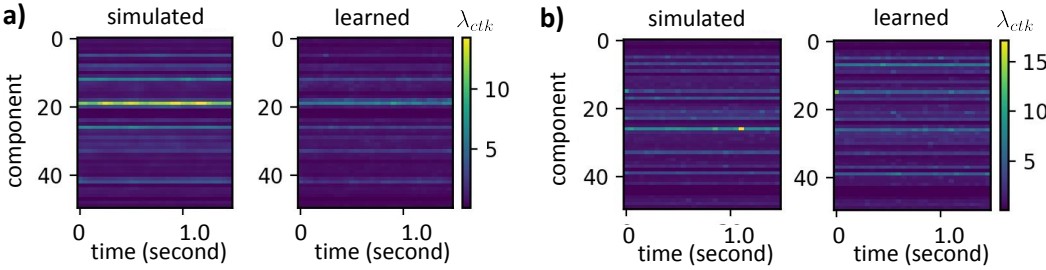

Figure 11: **Our encoding model recovers the relationship between the simulated spiking activity and the simulated behavior correlate.** Panel (a) shows a comparison of the simulated firing rates conditioned on the binary behavior variable with the learned firing rates by our encoding model. In Panel (b), we compare the simulated firing rates conditioned on the continuous behavior variable with the learned firing rates from our encoding model.

# 11 Relationship to spike sorting

We conducted experiments to investigate the biological interpretation of our MoG units and the correspondence between single cells identified by KS and our MoG units. The agreement matrix between "hard" KS spike assignments and "soft" MoG assignments is shown in Figure 12. We calculated the conditional probability of spikes belonging to each MoG component, given that these spikes belong to the corresponding KS unit. Notably, KS units with large amplitudes are less likely to be split into multiple Gaussian components. This shows a reasonable correspondence between the Gaussian components and the spike-sorted units.

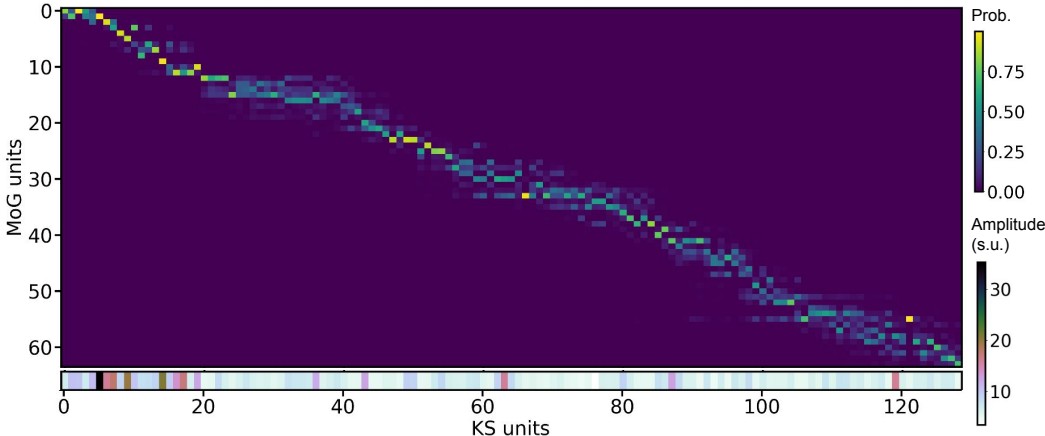

Figure 12: **Correspondence between Kilosort and MoG spike assignment**. Units are ordered by their depth on the Neuropixel probe. The color bar shows the conditional probability of spikes belonging to each MoG component, given that these spikes belong to the corresponding KS unit. The mean amplitude of each KS unit is shown at the bottom.

