# OpenReview forum: "Bypassing spike sorting: Density-based decoding using spike localization from dense multielectrode probes"
_NeurIPS.cc/2023/Conference — NeurIPS 2023 spotlight_

### Official Review · Reviewer_Cpfb · 2023-07-03

**Soundness:** 4 excellent
**Presentation:** 3 good
**Contribution:** 3 good
**Rating:** 8
**Confidence:** 3

**Summary:**

The authors demonstrate a novel approach to decoding behavioral variables from neural activity recording using dense silicon probes. Most approaches utilize a sort-then-decode line of attack – first channels are spike sorted, then decoders are trained on unambiguously identified single-units. Here, the authors present a probabilistic model that bypasses explicitly assigning waveforms to units; rather, the authors model spike features using a mixture of Gaussians, which is then leveraged for decoding. The authors go on to show that their method generally outperforms the current state-of-the-art in a variety of experiments spanning recordings in the mouse and in the monkey.

**Strengths:**

Strengths:

1. The authors present an elegant solution a highly-challenging problem in neuroscience – decoding behavior in the presence of noisy and highly-overlapping signals.
1. The results are generally compelling. Clearly, the author's algorithm outperforms Kilosort and multi-unit thresholding in a variety of decoding contexts.
1. The core idea and implementation is fairly straightforward. Given this, I expect this contribution to be generally useful to the community.

**Weaknesses:**

Weaknesses:

Major:

1. One potential issue with "bypassing" spike-sorting is noise and motion artifact. Spike-sorting, if carefully curated, can remove unwanted sources of contamination in extracellular electrophysiology. Here, I am somewhat concerned that this method could pick up and amplify contaminating signals that may contain behaviorally relevant information. For instance, Figure 3a shows that "good" KS units in CA1 and DG contain barely any information about motion energy; however, multi-unit thresholding, all KS, and the author's method find a surprising amount of information. Do the authors think this is reasonable? I am a bit worried since similar gains are not seen for choice decoding, which may not be as easy to decode from motion artifact. This could absolutely be real information uncovered by the model, but this issue is worth addressing head on. Do the authors think that their method could be picking up on any noise or motion artifact in the original recordings?

Minor:

1. Rhetorical point. Line 47 the authors mention prior attempts to "bypass" spike sorting as working with "limited" features such as maximum amplitudes and principal components of the spike waveform. Unless I missed something this language seems overly critical since the authors are only using peak-to-peak amplitudes and spike localization features (x, z).
1. Related to the last point, did the authors consider waveform principal components or aspects of the waveforms other than amplitude? Spike width, e.g., is known to contain information about cell type.
1. Line 97 typo

**Questions:**

1. Did the authors check to see if their technique picks up on noise or motion artifact in these recordings?

**Limitations:**

Limitations are adequately addressed.

---

> ### Author Rebuttal · Authors · 2023-08-09
>
> We thank reviewer Cpfb for their thorough and thoughtful review. We appreciate that the reviewer recognized the contribution of our method to the neuroscience community.
>
> **Major Weakness:**
> We performed motion correction (registration) during the preprocessing step, ensuring that motion artifacts are minimized in our data. Figure 5 in the supplementary materials provides a comparison of spike raster plots with and without registration, confirming the minimal impact of motion artifacts in the preprocessed data. Additionally, Kilosort has its own motion correction step, reducing the influence of motion artifacts on the spike-sorted decoder.
>
> Our decoder exhibits strong performance not only in decoding motion energy but also in decoding wheel speed and other continuous behaviors that are unrelated to motion artifacts. Decoding continuous behavioral variables is generally easier to improve compared to decoding binary choices. We acknowledge the importance of improving choice decoding, and we are actively exploring this direction in our ongoing research.
>
> The subpar performance of "good KS" is due to IBL's stringent quality control criteria for spike sorting. Only those single-units that pass the quality control procedure are categorized as "good" KS. In some brain regions like CA1 and DG, there are often less than 10 units that meet these criteria, explaining the poor decoding performance of "good" KS in these areas. This is precisely why decoding using all Kilosort units (including probable multi-unit activity) yields significantly better results.
>
> **Minor Weakness:**
> 1. We apologize for being overly critical with our choice of words. Our intention was to emphasize the importance of spike localization features, which are uniquely applicable to high-density probes with high spatial resolution. This novel technique, as described in this paper [(Boussard et al ‘21)](https://openreview.net/pdf?id=ohfi44BZPC4), is of great importance as it extracts valuable information that cannot be achieved solely through waveform features. Our ablation study in Section 8 of the supplementary materials supports this statement, showing that the inclusion of waveform features alongside localization features does not contribute significantly to the decoding tasks when compared to decoding solely based on localization features and peak-to-peak amplitudes.
> 2. We appreciate the insightful suggestion from the reviewer. We did explore the possibility of incorporating more waveform features to enhance decoding performance. As shown in our ablation study in Section 8 of the supplementary materials, we included waveform principal components in addition to localization features and peak-to-peak amplitudes. However, supplementary Table 3 clearly demonstrates that the addition of waveform features did not result in a significant improvement in the decoding tasks compared to using only localization features and peak-to-peak amplitudes.
> 3. We thank the reviewer for bringing this typo to our attention, and we will correct it in the future version.
>
> **Question 1:**
> We examined our method to ensure that it did not capture motion artifacts in the recordings. We manually inspected the spike raster plots after registration, and we found no evidence of motion artifacts affecting the decoding results. For a more comprehensive discussion on this topic, please refer to our response in the Major Weakness section.

---

> > ### Comment · Reviewer_Cpfb · 2023-08-13
> > **Response to author rebuttal**
> >
> > First, I'd like to thank the authors for the comprehensive rebuttal, I think they have done an excellent job of addressing the reviewers remaining concerns. I have already raised my score to strong accept.
> >
> > However, one thing that I'm still a little confused about re: motion artifacts: as the authors mention they apply drift correction, which corrects for the apparent physical displacement of units over time. I was not clear about this, but I was getting at a slightly different point. If I collect all spike waveforms that cross a very small threshold, say one standard deviation, many of the waveforms will likely arise from electrical and mechanical artifacts and not neurons. If I built a complex decoder using these waveforms I might be able to accurately decode movement – something correlated with waveforms that arise from artifacts – but not more cognitive variables such as choice.
> >
> > I tried to glean the inclusion criteria for spike waveforms that are subjected to modeling in the supplement, but it is still unclear what criteria a waveform must pass to be modeled in the first place. It's possible I missed this, but I could not find the exact details in the supplement. How were "candidate spike events" determined? Do the results depend on how this is parameterized at all? If I applied this method to a channel with only electrical noise and movement artifacts what would the result be?
> >
> > Also after looking over the supplement again it looks like some references are missing (e.g. 33 and 34, could be more).
> >
> > Thanks.

---

> > > ### Author Response · Authors · 2023-08-15
> > > **Response to reviewer Cpfb**
> > >
> > > We appreciate the kind words from reviewer Cpfb and thank them for their insightful questions. We did investigate the impact of motion artifacts on decoding accuracy, as evidenced in Figure 2 panel (d) of our main paper. This figure shows that the decoding quality of motion energy declines as the amount of motion artifacts in the data increases, which is the opposite of what we would expect if we are just decoding better because of noise artifacts. Additionally, to show that we are not performing better because we are finding additional spikes that a spike sorter would miss, we conducted an experiment. We fitted our model using only spikes detected by Kilosort 2.5, and compared its performance to decoders using spike-sorted outputs and our subtraction-based spike detection on choice and motion energy decoding. The results are summarized in the table below. As shown in the table, our decoder can achieve comparable or better decoding performance than the spike-sorted decoder when modeling the same spikes. This suggests that the gain in motion energy can be attributed to the density-based approach.
> > >
> > > |               | Density-based (subtraction spikes) | Density-based (KS spikes) |   Sorted (KS spikes)    |
> > > |---------------|------------------------------------|---------------------------|-------------------------|
> > > |     Choice    |           0.876 (0.068)            |       0.876 (0.079)       |       0.887 (0.078)     |
> > > | Motion energy |           0.589 (0.111)            |       0.579 (0.121)       |       0.503 (0.117)     |
> > >
> > > Regarding the inclusion criteria for spike waveforms, we use a subtraction-based spike detection as described in Section 4 of the supplementary materials. Specifically, For a series of voltage thresholds in standard units (12, 10, 8, 6, 5):
> > >
> > > 1. Detect threshold crossings;
> > > 2. Denoise using a neural network or TPCA;
> > > 3. Subtract and store spike events which would decrease the squared norm of the residual by at least 10 squared standard units when subtracted.
> > >
> > > While the subtraction-based detection procedure is a novel aspect of our method, it remains uncertain whether its parameterization affects decoding. We acknowledge the reviewer's point that the inclusion criteria for spikes may affect decoding accuracy to some extent. However, exploring different inclusion criteria would demand substantial effort, and we intend to perform additional experiments to confirm this aspect in the future.
> > >
> > > We apologize for the missing references and will include them in the future version of our paper.

---

> > > > ### Comment · Reviewer_Cpfb · 2023-08-15
> > > >
> > > > Thanks again to the authors for the fast response, they have gone above and beyond! This manuscript is definitely an exciting result for the field.

---

### Official Review · Reviewer_C56U · 2023-07-07

**Soundness:** 4 excellent
**Presentation:** 4 excellent
**Contribution:** 4 excellent
**Rating:** 8
**Confidence:** 4

**Summary:**

This paper presents a way to decode behavior from neural recordings, bypassing an explicitly spike sorting step. They model individual spikes as coming form a mixture of gaussian distribution, and the assignment probability to different mixtures is used instead of a 'hard' assignment of each spike to a cell. Robust performance of this approach is shown across a large dataset of Neuropixels recordings.

**Strengths:**

While spike sorting is thought to incorporate the biological priors (ie., the fact that spikes come from cells, which carry information in a neural circuit), and promote robust decoding, deviating from the prior by developing a "soft" spike sorting approach seems to have helped in this paper. This is surprising and very impactful.

Well written paper.

**Weaknesses:**

All the results use very high-density neuropixel recordings. However, it is not clear if the method is a general replacement to spike sorting across different arrays, etc.

The encoding model depends only on behavior. However, there is a large amount of behavior-independent variability (ex. attention states) that are not incorporated in the encoding model.

**Questions:**

1. Could the paper describe the method as "soft spike-sorting"? This would help place the method in the right context.

2. Spike sorting is necessary for biological interpretation of neural recordings. Does soft-spike sorting lend such interpretation?

3. If one doesn't do spike sorting, one could use local field potentials, or other bandpass signals to supplement threshold crossings. One should compare with all those features to claim the superiority of the method.

4. How crucial are the preprocessing steps? Were they developed on the same dataset to optimize spike sorting? If so then the results might be confounded.

5. What are the computational costs of this method compared to spike sorting?

6. Using the KS spike assignments, how 'pure' are the gaussians in the MOG? Are single cells split into different gaussians , or do single gaussians correspond to multiple cells? This could shed light on how important is it to deviate from "hard" spike sorting.

7. While spike sorting gives stable decoding across time (inspite of template changes over time), does the current method present with similar stability? Can you learn a decoder from the beginning of a recording and apply it towards the end?


**Limitations:**

Limitations have been adequately addressed.

Question for Area chair: The paper identified the source of the data as the data from International Brain Lab. It is not clear if the data is open-source. If not, then it clearly violates the double-blind policy of NeurIPS. Please check.

---

> ### Author Rebuttal · Authors · 2023-08-09
>
> We thank Review C56U for the thoughtful review and useful comments. We are glad that the reviewer felt that the manuscript was clearly written and appreciate that the reviewer recognizes that our work makes an important contribution to the literature.
>
> **Weakness 1:**
> Although our decoder is mainly designed for high-density (HD) probes, we acknowledge the reviewer’s point about adapting our decoder to more general probe geometries. The global response Table 1 demonstrates the effectiveness of our decoder across probe geometries beyond Neuropixel probes. Specifically, on both multiple electrodes and HD probes data, our decoder outperforms the previous clusterless decoder.
>
> **Weakness 2:**
> While our model is primarily conditioned on the behavior of interest, we have the flexibility to turn off this conditioning and utilize a vanilla GMM model to capture behavior-independent variability. This approach comes with a slight trade-off in decoding accuracy. For a detailed comparison between our models with and without dependencies on behavior correlates, please refer to Section 8 (ablation study) in the supplementary materials. We agree that including additional latent variables in the model (e.g. attention states) would be an exciting direction for future work.
>
> **Questions:**
> 1. Indeed, our method uses a GMM to estimate the probability of a spike belonging to a mixture component, thereby quantifying the uncertainty in spike assignment. This approach allows the model to make errors and retain valuable information, as opposed to "hard" spike sorting methods that discard such uncertainty.
> 2. While our method may not offer a single-unit interpretation similar to spike sorting, it brings its own advantages. Traditional spike sorting can be stringent and might discard valuable information by eliminating many cells that do not pass the quality control criteria. In contrast, our approach leverages all unsorted spikes within each brain region for decoding behavioral correlates. As a result, our method retains all available information within that region, providing abundant interpretations for specific brain regions without discarding any valuable data.
> 3. We agree with the reviewer's suggestion that the inclusion of LFP and other band-passed signals would be very informative. However, incorporating these data modalities would require extensive efforts beyond the scope of our current work. Our focus in this study is to understand what spikes can inform us about the behavior of interest, rather than optimizing decoding accuracies using all available data modalities. Nonetheless, we acknowledge the potential benefits of incorporating additional data modalities and consider it interesting future work.
> 4. No, the preprocessing steps were not developed to optimize spike sorting; they are distinct and separate processes. For instance, destriping is specifically employed for data from IBL due to prominent stripes in the recording data, and it is a preprocessing step developed by IBL. We rely on standard preprocessing from IBL to address quality concerns; see details in their white paper [(IBL et al ‘22)](https://figshare.com/articles/online_resource/Spike_sorting_pipeline_for_the_International_Brain_Laboratory/19705522). Additionally, registration, which removes probe motion, is crucial for data quality, and we utilized a separate method proposed in this paper [(Windolf et al ‘22)](https://www.biorxiv.org/content/10.1101/2022.12.04.519043v1). Spike localization is another essential step in our pipeline, and it is introduced separately in this work [(Boussard et al ‘21)](https://openreview.net/pdf?id=ohfi44BZPC4) as well.
> 5. In the global response, we provide a quantification of the computational cost for our method. Regarding Kilosort, we didn't run it ourselves and profile its computational cost, as the spike-sorted output is readily available from IBL's public database. It is worth noting that Kilosort is a real-time sorting algorithm.
> 6. Please refer to the section about Figure 3 in the global response for details.
> 7. While we can train a decoder at the start of a recording and apply it to the end, there could be a slight impact on decoding accuracy. We use spike features with registered locations that remain constant over time to mitigate probe drift or motion artifacts. The GMM components also remain unchanged over time. However, the mixing proportions of the GMM model vary over time. We conducted experiments by training the model on initial recording segments and then decoding subsequent segments for continuous behaviors on three datasets. The table below shows the mean correlation between true and decoded behaviors. In the context of this experiment, *Time shuffled* indicates training the GMM on segments from various time points within the recordings, while *Time ordered* indicates training on earlier segments to decode later ones.
>
> |              | Dataset 1 | Dataset 2 | Dataset 3 |
> |----------|----------|----------|----------|
> | Time shuffled | 0.760 (0.016) | 0.703 (0.015) | 0.802 (0.012) |
> | Time ordered | 0.733 (0.067) | 0.687 (0.008) | 0.792 (0.025) |
>
> **Question for Area Chair:**
> We use open-source IBL datasets in our paper. The public available datasets can be accessed [here](https://int-brain-lab.github.io/iblenv/public_docs/public_introduction.html).

---

### Official Review · Reviewer_Fmkq · 2023-07-07

**Soundness:** 3 good
**Presentation:** 3 good
**Contribution:** 2 fair
**Rating:** 4
**Confidence:** 4

**Summary:**

This paper proposed to decode animal behavior with a spike-sorting-free method that is well-suited for high-density recordings, by modeling the distribution of extracted spike features using a mixture of Gaussians (MoG) on uncertainty of spike assignments, and decoding using a generalized linear model (GLM). The authors benchmarked results on Neuropixel recordings from different brain regions, different animals, as well as different probe geometries, and showed that performance outperformed decoders based on multi-unit threshold crossings and single-units sorted by Kilosort.

**Strengths:**

1. The paper is clearly presented. The writings are easy to follow, and most of methods and results are described clearly.
2. Benchmarking results from multiple brain regions, animals, and different versions of Neuropixel probes were conducted, with a significant improvement on decoding accuracy compared to baseline methods.
3. Given that high-density recordings are getting more widely-used in the neuroscience field, this paper brings impacts with a better decoding tool for studies conducted with this kind of high-density probes.

**Weaknesses:**

1. While I recognize the paper providing extensive analysis results, the novelty of proposed density-based decoding on unsorted spike waveforms is limited from my perspective. First, the proposed algorithm on using MoG and GLM for decoding has been widely used on neural ensembles, or other types of neural signals like LFP or ECoG. For example, the [paper](https://www.ncbi.nlm.nih.gov/pmc/articles/PMC3972894/) also leveraged MoG for unsorted spike decoding, despite that it did leverage Bayesian decoding framework. This [paper](https://ieeexplore.ieee.org/stamp/stamp.jsp?tp=&arnumber=9585446) proposed Gaussian mixture of model (GMM)-assisted PLS (GMMPLS) for decoding in BMI. Second, there are also other methods proposed for decoding behavior from cluster-less neural data. For example this [paper](https://www.biorxiv.org/content/10.1101/760470v1.full.pdf) proposed cluster-less HMM model for decoding. Another [paper](https://www.biorxiv.org/content/biorxiv/early/2021/08/28/2021.08.26.457795.full.pdf) proposed Gaussian-process multi-class regression decoding on neural population data. The concept of modeling spike distributions with MoG has been applied, and GLM decoding also has been commonly used for neural ensemble. I recognized that some of these papers are not specifically targeting unsorted spikes, and acknowledge this paper bring good impacts for the neuroscience community, but I am not sure if NeurIPS is the proper target given novelty of its proposed algorithm. To meet the bar for NeurIPS, I would like to see more originality algorithm improvements.
2. Unclear whether baseline decoders are fair baselines. First, Kilsort depends on hyperparameters that need to tune properly for each dataset. From some of example decoded curves, I'm not sure if those baseline decoders were calibrated properly to data. In practice, we rarely worked with decoders with poor accuracy. This leads me to suspect the baseline decoders were not trained/tuned properly for comparison. Second, there are more SOTA decoders that have been used, including previous clusterless decoders on unsorted spiking data (see [paper](https://journals.physiology.org/doi/full/10.1152/jn.01046.2012)). Without a benchmark comparison to these decoders, it's hard to be convinced that the proposed algorithm is much superior than those other decoders.
3. Unclear if the proposed method only worked better for high-density recordings from Neuropixel or not. Despite that the authors provided two different probe versions for different geometries, it's unclear whether this brings similar benefits for other types of neural recordings (e.g., multiple tetrode arrays, etc). Especially if the paper claims that this method would be better than previous clusterless decoders tested on tetrode recordings, it's important for the NeurIPS audience to know if the proposed method works more generally on decoding unsorted spikes, or somehow better tuned for the distribution from Neuropixel recordings.

**Questions:**

1. I'm not following the authors' arguments on why previous Bayesian decoding methods on unsorted spikes do not apply for HD probes. Specifically, this argument "the aforementioned approaches are not suitable for HD probes as they rely on sampling-based Bayesian inference and are specifically designed for tetrodes". It's true that sampling-based Bayesian inference makes certain assumptions on point process, but I don't quite see why those are specific for tetrodes. The original Bayesian inference paper used waveform features such as the maximum amplitudes and principal components for decoding, but that do not mean waveform features cannot be represented in other formats. Can authors elaborate this statement?
2. What are the baseline decoders used with sorted neurons after sorted by Kilsort?

---

> ### Author Rebuttal · Authors · 2023-08-09
>
> We thank reviewer Fmkq for their thorough and thoughtful review. We appreciate their valuable feedback and would like to address their main concerns.
>
> **Weakness 1:**
> Please refer to the global response for the novelty of our method.
>
> **Weakness 2:**
> It is important to clarify that the baseline decoders are well-tuned and calibrated properly for each dataset, and we used spike-sorted data tuned by IBL, as described in IBL's spike sorting white paper [(IBL et al ‘22)](https://figshare.com/articles/online_resource/Spike_sorting_pipeline_for_the_International_Brain_Laboratory/19705522). The relatively lower decoding accuracy is due to the inherent difficulty of the decoding task with the behavior data from IBL, which are challenging to decode, especially considering that some brain regions contain limited information about the behaviors. In fact, in the IBL's decoding paper titled "*A Brain-Wide Map of Neural Activity during Complex Behaviour* [(IBL et al ‘23)](https://www.biorxiv.org/content/10.1101/2023.07.04.547681v2)," their achieved decoding accuracies are comparable to ours. For detailed numerical values, please consult their Figure S6 and S15 in the supplementary materials. Additionally, in the global response Table 1, we showcase high decoding accuracies for multiple tetrodes data, which involves a simpler decoding task compared to the IBL's behavior tasks.
>
> We acknowledge the importance of conducting a benchmark comparison of clusterless decoders and appreciate the reviewer for bringing up previous clusterless decoding papers. It is worth noting that the Gaussian-process multi-class regression decoding paper [(Greenidge et al ‘21)](https://www.biorxiv.org/content/biorxiv/early/2021/08/28/2021.08.26.457795.full.pdf) and the GMM-PLS paper [(Foodeh et al ‘21)](https://ieeexplore.ieee.org/stamp/stamp.jsp?tp=&arnumber=9585446) are not specifically designed for clusterless decoding, hence we did not consider comparing our method to those. Regarding a comparison with the previous clusterless decoders, please refer to the global response for details.
>
> **Weakness 3:**
> We acknowledge the reviewer’s point about adapting our decoder to more general probe geometries. The global response Table 1 demonstrates the effectiveness of our decoder across probe geometries beyond Neuropixel probes. Specifically, on both multiple tetrodes and HD probe data, our decoder outperforms the clusterless point process decoder [(Denovellis et al ‘21)](https://elifesciences.org/articles/64505). It is important to note that our method is not exclusively tailored for Neuropixel recordings, and can be applied more generally to probes with different geometries.
>
> **Question 1:**
> The key point to emphasize is the high data volume associated with HD probes, which generates a significantly larger amount of data compared to tetrodes. As a result, previous clusterless decoders based on sampling-based inference tend to be slow. This computational cost is evident in the clusterless HMM [(Ackermann et al ‘19)](https://www.biorxiv.org/content/10.1101/760470v1.full.pdf) method, as mentioned by the authors in their paper. They admitted the challenges related to high computational cost, with the model taking about 6 hours to fit a relatively small dataset. Although the clusterless point process model [(Denovellis et al ‘21)](https://elifesciences.org/articles/64505) has a lower computational cost, our decoder is faster and outperforms it in high-density settings, as indicated in global response Table 1 and Figure 2. The limitation of their approach lies in its inability to effectively handle the highly overlapping signals of HD probes, where multiple electrodes are closely packed together. Moreover, our density-based decoder is more flexible by avoiding making restrictive assumption about the underlying system dynamics.
>
> The original method could potentially use features of other formats, but it's important to note that the localization features, which are HD-specific, play a crucial role. HD probes offer exceptional spatial resolution, and utilizing spike localization features is essential for successful decoding. Relying solely on waveform features on each electrode becomes challenging in the context of HD probes without leveraging the spatial advantages offered by localization features. Our ablation study in Section 8 of the supplementary materials supports this statement, showing that the inclusion of waveform features alongside localization features does not contribute significantly to the decoding tasks when compared to decoding solely based on localization features and peak-to-peak amplitudes.
>
>
> **Question 2:**
> We use ridge regression for decoding continuous behaviors and L2-penalized logistic regression for decoding binary behaviors.

---

### Official Review · Reviewer_tDhN · 2023-07-12

**Soundness:** 3 good
**Presentation:** 3 good
**Contribution:** 3 good
**Rating:** 7
**Confidence:** 4

**Summary:**

The paper develops a decoding method directly on ‘spike features’, without going through spike sorting. The authors model spike assignment uncertainty using a mixture of Gaussians model, and then perform variational inference to model the relationship between the spike features and behavior.

**Strengths:**

The premise and models in this paper are sound, if not particularly novel. The inference methods are not particularly novel either. However, it is clearly shown that the models developed here are empirically quite powerful. The evaluations performed are very extensive, showing very clearly that their methods outperform standard spike sorting and decoding methods on large datasets in multiple species.

**Weaknesses:**

It is unclear why exactly previous clusterless methods fail on this kind of data. The authors may need to show this concretely to be fair to previous approaches that do not use spike sorting.

Simulation data may help to get intuition on the methodology, and to show validation of the methods.

**Questions:**

It may be a good idea to apply previously developed clusterless methods on this kind of data to show how / why they fail.

Simulated data will be helpful to the reader to better understand the limitations of the modeling approach.

**Limitations:**

The limitations are adequately discussed. Potential negative societal impact is adequately discussed.

---

> ### Author Rebuttal · Authors · 2023-08-09
>
> **General:**
> We thank Reviewer tDhN for carefully reviewing our manuscript, and appreciate the opportunity to address the concerns raised. We want to emphasize the novelty of our method, which involves conditioning of the data generating process on external variables, allowing for improved prediction. Our approach is not limited to GMM modeling and GLM decoding; instead, it can be extended to non-Gaussian mixture models and nonlinear models to tackle diverse modeling challenges. Furthermore, compared to previous clusterless decoders based on state-space models, our density-based decoder avoids making explicit assumptions about underlying system dynamics, and is thus more flexible at capturing complex relationships in the data. Our method is also more scalable than previous approaches, which is desirable for large data volumes generated by HD probes.
>
> **Weakness / Question 1:**
> We have compared our decoder to previous clusterless decoders on both multiple electrodes and HD probes to demonstrate that our method is effective in both scenarios. The global response Table 1 provides a benchmark of clusterless decoders, reaffirming the superiority of our density-based decoder when applied to HD probes.
>
> **Weakness / Question 2:**
> We appreciate the reviewer's feedback on conducting a simulation study for model validation. The simulation results shown in global response Figure 1 confirm that our encoding model effectively captures the association between the simulated spike features and the simulated behavior of interest. With such learned associations, the decoding model is able to accurately decode the behaviors. To highlight the significance of these learned associations, we have compared our GMM conditioned on the decoding variable with a vanilla GMM that lacks such informative associations. The details of this comparison can be found in Table 2 in Section 8 (ablation study) of the supplementary materials.

---

> > ### Comment · Reviewer_tDhN · 2023-08-16
> > **Response to Rebuttal**
> >
> > I want to thank the authors for their detailed and comprehensive response. My questions have been adequately answered and the new results are reasonable. Although a little low on novelty, the paper fills a valuable gap in a fast-expanding field. I am accordingly raising my score.

---

### Author Rebuttal · Authors · 2023-08-09

We appreciate the reviewer's feedback on our manuscript. We were encouraged that the reviewers recognized our paper's empirical robustness and its solution to the challenging task of decoding behaviors from highly overlapping neural signals. To address the concerns, we conducted the following experiments:
- Benchmark comparison to other clusterless decoders.
- Simulation for model validation.
- Computational time cost comparison.
- Correspondence between spike sorting and GMM assignment.

One important point to emphasize is the novelty of our method, which involves conditioning the generative model for the spike feature distribution on the behavior of interest, making the inference problem non-trivial. See supplementary materials for detailed derivations. **The probabilistic model and inference method presented have broader relevance within the NeurIPS community. The conditioning of the data generating process on external variables allows for improved prediction in various scenarios.  Our approach is not limited to GMM and GLM; instead, it can accommodate non-Gaussian mixture models and nonlinear models.** While our paper serves as a proof-of-concept, the proposed method can be extended to tackle diverse modeling challenges.

**Our novelty also lies in the first demonstration of a clusterless decoding method designed for high-density (HD) probes (which are now used in hundreds of labs), and our use of localization features which are only meaningful for HD probes.** By combining these factors, we overcome previous challenges and improve decoding performance. Furthermore, compared to previous decoders based on state-space models, our density-based decoder offers increased flexibility and scalability. Our method avoids explicit assumptions about underlying system dynamics, making it more flexible at capturing complex relationships in the data. The increased scalability is critical for large data volumes generated by HD probes.

We appreciate the suggestion to benchmark clusterless decoders beyond HD probes. However, limited available code in clusterless decoding papers makes comparisons challenging within a short timeframe. While the suggested Bayesian decoding paper [(Chen et al ‘12)](https://www.ncbi.nlm.nih.gov/pmc/articles/PMC3972894/) lacks functionality for the main task (due to incomplete code), we re-implemented the clusterless HMM model [(Ackermann et al ‘19)](https://www.biorxiv.org/content/10.1101/760470v1.full.pdf), but the training time was prohibitive. Despite these limitations, we made efforts to compare our method to clusterless point process decoders [(Denovellis et al ‘21)](https://elifesciences.org/articles/64505) on both tetrodes and HD probes. This clusterless point process decoder is similar to the Bayesian decoding paper, as both use a marked point process to link spike features and behaviors. We used the same spike features and calibrated each decoder to the data. For multiple-tetrode data, both decoders used spike amplitudes from 4 channels of 5 tetrodes. Similarly, for HD probes, both decoders relied on spike localization features and peak-to-peak amplitudes. **Our method's advantage over the clusterless point process decoder is evident in Table 1,  owing to the increased flexibility of our decoder compared to earlier clusterless state-space models.**

We conducted simulations to illustrate the principles of our method. The simulation aimed to show that our encoding model can learn the relationship between spike features and behaviors. We performed two tasks, decoding a binary variable, $y_k$, simulated from a Bernoulli distribution, and decoding a continuous variable, $y_k$, simulated from a Gaussian process. To mimic the data-generating process, we selected Gaussian components with "templates" extracted from a real dataset. The encoding model parameters, $b$ and $\beta$, were also taken from learned parameters in the same dataset. Given $b, \beta$, and $y_k$, we simulated the "firing rates" $\lambda$ for each Gaussian component in the mixture, as described in the Method section of our paper. Next, we generated spike features based on these simulated ''firing rates'', and applied the encoding model to infer the behavior-dependent $\lambda$. Figure 1 displays the learned $\lambda$ for each component $c$, time $t$, and trial $k$. **The learned "firing rates" $\lambda$ closely resembled the simulated ones, indicating the model's ability to recover the primary associations between spike features and behaviors. With such associations, the decoding model can decode behaviors.**

We appreciate the question about the computational cost of our method. For spike sorting, we did not personally run Kilosort (KS), given the accessible spike-sorted output from IBL's public database. Notably, Kilosort operates in close to real-time, implying that it takes 1000 seconds to sort a 1000-second recording. In Figure 2, we provided a computational time comparison relative to real-time. **Our decoding step operates at a sub-real-time pace (0.3 times real-time), which is 4 times faster than the point process decoder (1.2 times real-time). The total time after preprocessing for our method is close to real-time.**  While we didn't measure the time cost of the clusterless HMM due to the extensive model fitting time, the authors acknowledged that their model takes around 6 hours to fit on a small dataset in the paper.

We appreciate the inquiry about the insights our method provides into "hard" spike sorting. In response, we computed an agreement matrix (Figure 3) between "hard" KS assignments and "soft" GMM assignments. We calculated the conditional probability of spikes belonging to each GMM component, given that these spikes belong to the corresponding KS unit. Notably, KS units with large amplitudes are less likely to be split into multiple Gaussian components. In conclusion, **Figure 3 shows a reasonable correspondence between the Gaussian components and the spike-sorted units.**

---

### Decision · Program_Chairs · 2023-09-21

**Decision:**

Accept (spotlight)

**Comment:**

The reviewers largely agree that this is an interesting and useful piece of work. Their opinions have been strengthened by the author responses, and the additional results/clarification provided. Please be sure to include these in the final version of the manuscript.